



# Influence of air mass origin on microphysical properties of low-level clouds in a subarctic environment.

Konstantinos M. Doulgeris[1], Ville Vakkari[1,2], Ewan J. O'Connor[1], Veli-Matti Kerminen[4]
Heikki Lihavainen[1,3] and David Brus[1]

[1]Finnish Meteorological Institute, Erik Palménin aukio 1, P.O. Box 503, FIN-00100 Helsinki, Finland
[2]Atmospheric Chemistry Research Group, Chemical Resource Beneficiation, North-West University, Potchefstroom, South Africa
[3]Svalbard Integrated Arctic Earth Observing System (SIOS), SIOS Knowledge Centre, Svalbard Science Centre, P.O. Box 156, 9171 Longyearbyen, Norway
[4]Institute for Atmospheric and Earth System Research/Physics, Faculty of Science, University of Helsinki, Helsinki, Finland

*Correspondence to*: K.D (konstantinos.doulgeris@fmi.fi)

**Abstract:** In this work, an analysis was performed to investigate how different long-range transport air masses can affect the occurrence and the microphysical properties of low-level clouds in a

clean subarctic environment. The cloud measurements comprised in-situ and ground-based and were conducted during eight Pallas Cloud Experiments (PaCE) held in the autumn between 2004 and 2019. Each PaCE was carried out at the Pallas Atmosphere – Ecosystem Supersite, located in the Finnish subarctic region. Two cloud spectrometer ground setups were installed on the roof of the station to measure cloud microphysical properties: the Cloud, Aerosol and Precipitation Spectrometer probe

(CAPS) and the Forward Scattering Spectrometer Probe (FSSP). Air mass histories were analyzed using the Lagrangian particle dispersion model FLEXPART in order to investigate the differences between five distinct source regions (Arctic, Eastern, Southern, Western and Local). The results showed that there was a connection between the microphysical properties of the clouds we sampled during PaCEs and the air mass source region. The occurrence of clouds at the measuring station was mainly connected

with air masses coming from the continental regions. Higher values of cloud droplet number concentrations ($N_c$) were related to air masses coming from the Southern and Eastern regions (continental), whereas the lowest values of $N_c$ were related to Arctic and Western air masses (marine). In addition, the marine air masses were characterized by larger cloud droplets (median volume diameter approximately from 15 to 20 µm), in comparison to continental air masses that were mainly

characterized by cloud droplets with median volume diameters of approximately from 8 to 12 µm. Finally, there was an indication that cloud droplets were more prone to grow in warm liquid clouds.




## 1. Introduction

Uncertainties in cloud processes and feedbacks are key challenges when developing climate projections (e.g., Boucher et al., 2013; Sherwood et al., 2020). Cloud microphysics and their dynamics are considered as a fundamental challenge (Morrison et al., 2020) due to their connection to the cloud radiative effect (e.g., Devenish et al., 2012; McFarquhar et al., 2020). Thus, it is necessary to distinguish between the effects of aerosol and varying meteorological conditions on clouds (Barthlott and Hoose,

2018) since the aerosol is influenced through meteorology by air mass history as well as cloud and precipitation processes (Rosenfeld et al., 2014). Long-range transport is significant when investigating the characteristics and the spatial distribution of aerosols (e.g., Raatz and Shaw 1984; Barrie 1986; Freud et al., 2017; Wang et al., 2020; Lee et al., 2022). Due to insufficient knowledge of the cloud dynamics and the interaction between aerosols and clouds (e.g., Painemal et al., 2014; Orbe et al.,

2015a; Fuchs et al., 2017; Cho et al., 2021), it is important to understand how different air masses can influence the aerosols and the cloud microphysics. Investigating subarctic clouds is of particularly high interest due to the Arctic amplification effect, since the Arctic surface energy budget and Arctic warming feedback are affected by cloud related radiative processes (e.g., Wendisch et al., 2019, Shupe at al.,2022).

Several observation efforts and experiments have been made to explore how air masses affect climate and the cloud macrophysical and microphysical properties (e.g., Hobbs and Rangno 1998; Gultepe et al., 2000; Orbe et al.,2015b; Solomon and Shupe 2019; Delgado et al., 2021). Hobbs and Rangno (1998) highlighted that air masses from the south resulted in the highest overall aerosol number concentration measured in altocumulus clouds over the Beaufort Sea. Gultepe et al. (2000) stated that

Arctic clouds were affected by the air mass origin, which was strongly related to aerosol properties, dynamical and thermodynamical parameters. Gultepe and Isaac (2002) studied the cloud microphysics over the Arctic Ocean and found that there were differences in the number concentration, liquid water content and effective radius of Arctic clouds in air masses originating from the Arctic and Pacific Oceans. After investigating the air mass origin seasonality Orbe et al. (2015a) revealed that the northern

hemisphere summer air-mass origin response to increases in greenhouse gases (Orbe et al., 2015b). Fuchs et al., (2017) highlighted the impact of air mass origin and dynamics on cloud property changes in the Southeast Atlantic during the biomass burning season based on a cluster analysis of 8 years of September data. Solomon and Shupe (2019) presented a case study of a sharp transition between high ice clouds and the formation of lower stratocumulus from Summit, Greenland, when a warm and moist

air mass was advected to Greenland from lower latitudes. Iwamoto et al. (2021), using measurements from a high mountain site located in Japan, showed that the cloud droplet number concentrations were significantly higher in continental air masses than in air masses from the Pacific Ocean. Patel and Jiang (2021) combined measurements of aerosol properties from a site located in Lamont, Oklahoma, with cluster analysis of back trajectories to study aerosol characteristics and their influences on cloud



condensation nuclei (CCN) under various air mass environments and suggested that information on the aerosol chemical composition and mixing state are more crucial at lower supersaturations. Delgado et al. (2021), using aerosol and cloud measurements from a site in a tropical montane cloud forest on the Caribbean Island of Puerto Rico, suggested that air masses that arrived after passing over areas with anthropogenic emissions led to clouds with much higher cloud droplet concentrations. Cho et al. (2021)

investigated wintertime cloud properties and radiative effects in connection with cold and warm air mass origins at Ny-Ålesund, Svalbard, using remote sensing measurements with cloud radar, ceilometer and microwave radiometer, and revealed that the effective radius of ice particles in warm advection cases was approximately 5–10 µm larger than that of cold advection cases at all altitudes.

One of the few sites that enables long-term in situ observations of cloud and aerosol properties

in arctic and subarctic air masses is the Pallas Global Atmospheric Watch (GAW) station in northern Finland (e.g., Lihavainen et al., 2008; Hyvärinen et al., 2011; Anttila et al., 2012; Raatikainen et al., 2015; Gérard et al., 2019; Girdwood et al., 2020, 2022). However, after the initial case study (Lihavainen et al., 2008), which indicated a clear Twomey effect depending on air mass origin, no subsequent concerted study has investigated the effect of air mass origin on cloud microphysical

properties at Pallas. Here, we focus on quantifying the impact of air mass origin (e.g., clean arctic vs. long-range transported air from continental Europe) on low-level clouds and their patterns based on measurements at the Pallas GAW station. Size distribution is considered as one of the most important parameters of the cloud system due to its impact on the dynamics and microstructures within the cloud (Igel et al., 2017a, b). Measuring cloud microphysical properties, such as the median volume diameter

and liquid water content, is of high importance for the identification and description of clouds (Pruppacher and Klett, 1977; Rosenfeld and Ulbrich, 2003; Donovan et.al., 2015), cloud radiative properties and lifetime (Albrecht 1989; Small et al., 2009;), and the probability for which clouds precipitate (Rosenfeld and Ulbrich, 2003; Chang et al., 2019). We used *in situ* low-level cloud measurements from two ground-based cloud spectrometers from eight different years of campaigns to

obtain the cloud droplet size distribution. The FLEXible PARTicle (Flexpart) dispersion model was used to analyze the air mass history. A description of the measurement site and the instrumentation, as it was installed, is given in Sect. 2.1 and 2.2. Subsequently, in Sect. 2.3, a general overview of the campaigns is presented. In Sect. 2.4, we present how the backward trajectories were calculated. In Sect. 3, the optimal threshold of travelling air masses within a region to represent an air mass type is

identified, and a detailed analysis is made to find out to what extent the air mass type influences the microphysical properties of the low-level clouds. Finally, in Sect. 4, we summarize our main conclusions.




## 2. Methodology

### 2.1 Sampling Station


The measurements were conducted in subarctic Finland at the Pallas Atmosphere – Ecosystem Supersite (67°580 N, 24°070 E), hosted by the Finnish Meteorological Institute. The site where the ground-based cloud spectrometers were installed was the Sammaltunturi station, located on a hilltop, 565 m above sea level (ASL) (Hatakka et al., 2003). The site where the ceilometer was installed was the Kenttärova

station, 347 m ASL, located at the foot of the same hill (Fig.1). A full description of the Pallas Atmosphere – Ecosystem Supersite can be found in Lohila et al. (2015).

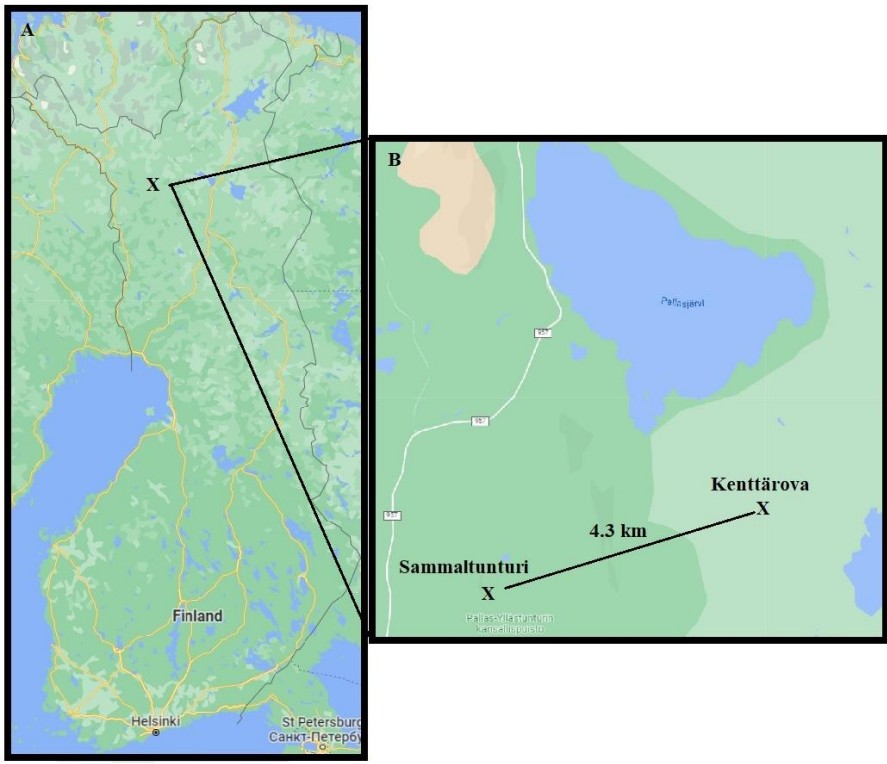

Figure 1. a) Map of Finland showing the location of the Pallas Atmosphere – Ecosystem Supersite (black cross), and (b) map of the wider Pallas area showing the location of Sammaltunturi and

Kenttärova (black crosses) (© Google Maps).

### 2.2 Instrumentation

The instruments that were used in this study are listed in Table 1, together with the measured and derived

parameters, their uncertainties and their location. During PaCEs, we used ground-based in situ cloud





spectrometers to monitor the cloud droplet size distribution, which has been recognized as a valid method for continuous cloud in situ measurements in the ACTRIS network (Wandinger et al., 2018). Four microphysical parameters were derived from the measured size distribution (Droplet Measurement Technologies Manual, 2009; Doulgeris et al., 2020): the cloud droplet number concentration ($N_c$, cm$^{-3}$), the median volume diameter (MVD, µm) and effective diameter (ED, µm) of cloud droplets, and the cloud liquid water content (LWC, g m$^{-3}$).


Two ground-based spectrometers were used: the cloud, aerosol and precipitation spectrometer (CAPS) and the forward-scattering spectrometer probe (FSSP-100, hereafter called FSSP for simplicity) (Fig. 2). CAPS was made by Droplet Measurement Technologies (DMT), Boulder, CO, USA. FSSP-100 (model SPP-100, DMT) was initially manufactured by Particle Measuring Systems (PMS Inc., Boulder CO, USA) and later acquired by DMT. The CAPS probe includes three instruments: the cloud and aerosol spectrometer (CAS), the cloud imaging probe (CIP), and the hot-wire liquid water content (LWC$_{hw}$) sensor. The size range of the CAS extends from 0.51 to 50 µm and that of the FSSP from 0.5 to 47 µm in diameter. In both CAPS and FSSP, the main measurement principle for the size detection is based on a conversion of the forward scattering of light into a size bin using Lorentz–Mie theory (Mie, 1908). Their main difference was that the CAPS was fixed and always heading to the main wind direction of the station (southwest, ∼ 225), whereas the FSSP-100 was deployed on a rotating platform to continuously face the wind. A description of both ground setups, installation, limitations and the methodology that was used is documented in Doulgeris et al. (2020) and Doulgeris et al. (2022). The instrument that monitored the cloud base was a lidar ceilometer (model CT25K, Vaisala Oyj, Vaisala users guide, 2002; Emeis et al., 2004), except in 2019 when it was replaced by a model CL31, Vaisala Oyj. The meteorological variables were monitored by an automatic weather station (model Milos 500, Vaisala Oyj). All the weather sensors that were used in this work were described in Hatakka et al. (2003). The temperature was measured at 570 m ASL by a PT100 sensor, the horizontal visibility by a weather sensor (model FDP12P, Vaisala Oyj), the relative humidity by a HUMICAP (Vaisala Oyj), the barometric pressure by a BAROCAP (Vaisala Oyj) sensors, the wind direction by a heated wind vane and the wind speed by a heated cup (Vaisala Oyj).





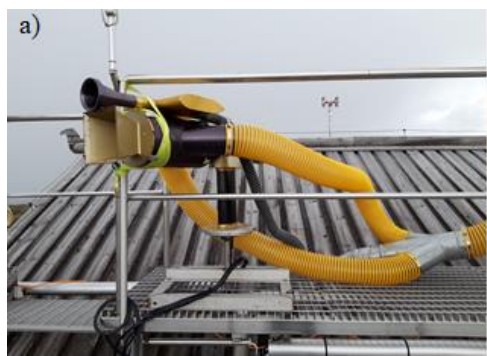
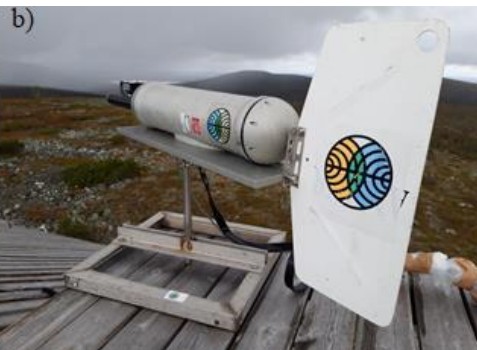



Figure 2. a) CAPS and b) FSSP-100 ground setups as installed on the roof of Sammaltunturi station
during PaCE 2017.

TABLE 1: Instrumentation that was used during PaCE along with measured and derived parameters,
their sampling frequencies, accuracy of the instruments and the location that were installed.

| Instrument | Measured, derived parameters | Sampling frequency | Accuracy | Location | References |
|---|---|---|---|---|---|
| CAS, DMT | Number size distribution of cloud droplets (0.51 to 50 µm); derived parameters $N_C$, LWC, ED, MVD | 1s | at ambient droplet concentrations of 500 cm$^{-3}$, 27 % undercounting 20 %–30 % oversizing | Sammaltunturi | Doulgeris et al., 2020, 2022. Baumgardner 2001. Lance (2012) |
| FSSP, DMT | Number size distribution of cloud droplets (0.5 to 47 µm); derived parameters $N_C$, LWC, ED, MVD | 1s | Nc accuracy: 16 % sizing accuracy: ±3 µm LWC accuracy: 30 %–50 % | Sammaltunturi | Doulgeris et al., 2020, 2022 Brenguier 1989 Baumgardner et al. (2017) Baumgardner (1983) |
| Ceilometer CT25K, Vaisala | Cloud base altitude | 60s | ±2 % ±1/2× (resolution) | Kenttärova | Vaisala users guide, 2002, Emeis et al., 2004 |
| FD12P, Vaisala | Horizontal visibility | 60s | ±10 % at 10 – 10.000 (m) | Sammaltunturi | Hatakka et., al 2003 |
| PT100 sensor, Vaisala | Temperature | 60s | ±0.1(ºC) | Sammaltunturi | Hatakka et., al 2003 |
| HUMICAP sensor, Vaisala | Relative humidity | 60s | ±0.8(%) RH | Sammaltunturi | Hatakka et., al 2003 |
| BAROCAP sensor, Vaisala | Barometric pressure | 60s | ±0.15(%) (hPa) | Sammaltunturi | Hatakka et., al 2003 |
| Wind vane, Vaisala | Wind direction | 60s | ±3 (º) | Sammaltunturi | Hatakka et., al 2003 |
| Heated cup, Vaisala | Wind speed | 60s | ±0.17 (ms$^{-1}$) | Sammaltunturi | Hatakka et., al 2003 |




### 2.3 Sampling campaigns

Measurements used in this study were conducted during the Pallas Cloud Experiments (PaCE). A description of the dataset (microphysical properties of clouds along with meteorological variables) that was obtained during PaCEs are available in Doulgeris et al. (2022). The PaCEs were approximately two-month-long field campaigns conducted in the Finnish subarctic region at the Sammaltunturi station during autumn and lasted approximately from the beginning of September until the end of November. The reason for this choice was that during autumn the Sammaltunturi station is frequently inside a cloud, which allowed us to perform ground-based, continuous in situ cloud measurements (Hatakka et al., 2003). An overview of each campaign along with the availability of instruments and the hours of observations in cloud are presented in Table 2. During PaCEs, all measurements were performed with a 1 Hz acquisition frequency. For the data analysis, averages per minute from each instrument were calculated when the measuring site was inside a cloud. Each cloud event was inspected separately. Afterwards, in-situ cloud data were related to the air mass origin and classified accordingly. We only used measurements when the cloud spectrometers were facing the wind direction, as suggested by Doulgeris et al. (2020).

Fine particles at the Sammaltunturi are expected to be dominated by sulphate and particulate organic matter in continental air masses, while particulate organic matter, sodium and chlorine are the main components in marine air masses (Lihavainen et al., 2008, Brus et al., 2013a). Also, episodes of elevated concentrations of $SO_2$ and $H_2SO_4$ are possible in air masses arriving from the Kola Peninsula, which is a large source of $SO_2$ emissions (Kyro et al., 2014; Sipilä et al., 2020, Brus et al., 2013a, b). Elevated $SO_2$ concentrations and particle number concentrations in the accumulation-mode (0.1–1 μm in diameter) of the mass size distribution are also expected from air masses travelled over continental Europe. Total aerosol particle number concentrations at Sammaltunturi are typically low (average of 700 $cm^{-3}$, in winter the daily averages may drop below 100 $cm^{-3}$) (Hatakka et al., 2003; Komppula et al., 2003). Generally, in Finnish Lapland, aerosol particle number concentrations are expected to be the highest in air masses arriving from the Kola Peninsula (more than 1000 $cm^{-3}$) and the lowest in marine air masses, especially in air originating from the Arctic Sea (often less than 100 $cm^{-3}$) (Lihavainen et al., 2008). Higher particle number concentrations in the accumulation mode are also expected in air masses which have travelled over the continental Europe (Virkula et al., 1997). In Sammaltunturi, cloud condensation nuclei (CCN) concentrations are smaller than 100 $cm^{-3}$ for supersaturations from 0.1 to 0.5%. The aerosol particle population is dominated by the Aitken mode (30–100 nm in diameter) and a low hygroscopicity is expected (Paramonov et al., 2015).

TABLE 2: Overview of each campaign, including the starting and ending date, the availability of the ground cloud spectrometer probes and the ceilometer model.





| Year | Starting date | Ending date | CAS | FSSP | Ceilometer | Cloud Observations CAPS (hours) | Cloud Observations FSSP (hours) |
|---|---|---|---|---|---|---|---|
| 2004 | 25.10. | 07.11. | Not available | On site | CT25K | - | 42 |
| 2005 | 30.09. | 05.10. | Not available | On site | CT25K | - | 45.4 |
| 2009 | 11.09. | 09.10. | Not available | On site | CT25K | - | 34.2 |
| 2012 | 14.09. | 30.10. | On site | On site from 09.10 | CT25K | 477.5 | 50 |
| 2013 | 14.09. | 28.11. | On site from 15.10 | On site | CT25K | 483.5 | 492.6 |
| 2015 | 24.09. | 02.12. | On site from 06.10 | On site | CT25K | 528.4 | 561.9 |
| 2017 | 18.09. | 29.11. | On site | Not available | CT25K | 681.8 | - |
| 2019 | 20.09. | 24.11. | On site | Not available | CL31 | 479.6 | - |

**2.4 Classification of air mass origin**

Air mass origins were analyzed using the Lagrangian particle dispersion model FLEXPART version 10.4 (Seibert and Frank, 2004; Stohl et al., 2005; Pisso et al., 2019). FLEXPART was run backward in time to calculate potential emission sensitivity (PES) fields. PES in a particular grid cell is proportional to the air mass residence time in that cell and was calculated in units of seconds (Seibert and Frank, 2004; Pisso et al., 2019). ERA5 reanalysis by European Centre for Medium-Range Weather Forecasts

(ECMWF) was used as meteorological input fields for FLEXPART at 1 hour temporal resolution and 0.25° resolution in latitude and longitude. In vertical, ERA5 levels 50 to 137 were used, which corresponds approximately to the lowest 20 km above surface. The model domain was from 125° W to 75° E and 10° N to 85° N, which was large enough to contain 96 h simulations backward in time. FLEXPART runs were initiated at an hourly time resolution for the in-cloud measurement periods at

Sammaltunturi. The retro plume release height was set to 560-660 m ASL, as the terrain height in ERA5 at the site was approximately 300 m ASL. The PES output resolution was set to 0.2° latitude and longitude with a 250-m height resolution up to 5 km and two additional output levels at 10 km and 50 km.

The air mass source regions for the Sammaltunturi site were divided into five categories: Arctic,

Eastern, Southern, Western and Local (Fig. 3), as proposed by Asmi et al. (2011). The Arctic and Western regions represent marine areas, whereas the Eastern, Southern and Local regions represent



continental areas. Figure 5a illustrates an example case for air masses arriving on 20.09.2012 at 11:00 UTC at Sammaltunturi. In Fig. 4a PES is summed up for the full duration of the 96-hour backward simulation and for all output heights at each latitude-longitude grid cell. Fig. 4b displays the vertical

distribution of PES during the simulation. However, the information in Fig. 4b is not used in further analysis, but fraction of PES in each source region is calculated based on the integrated data presented in Fig. 4a. For the case in Fig. 4a, this results in PES fractions of 42% Local (area V), 35% Southern (area III) and 23% Western (area IV). Finally, Fig. 4c shows how the PES fractions evolve during 20.09.2012 at Sammaltunturi with the case in Fig. 4a represented by the bar at 11 UTC. In Fig. 4c a

clear change in air mass origin is observed at 18 UTC when the fraction of Arctic source region starts to increase, reaching up to 48% at 22 UTC.

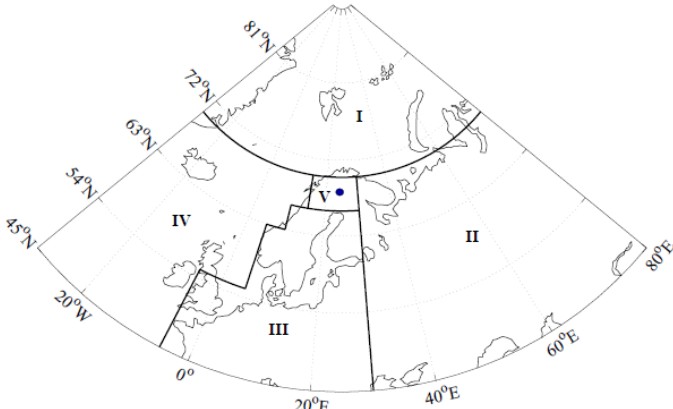

Figure 3: Map of the air mass regions: I (Arctic), II (Eastern), III (Southern), IV (Western) and V (Local). Figure was adopted from Asmi et al., 2011.


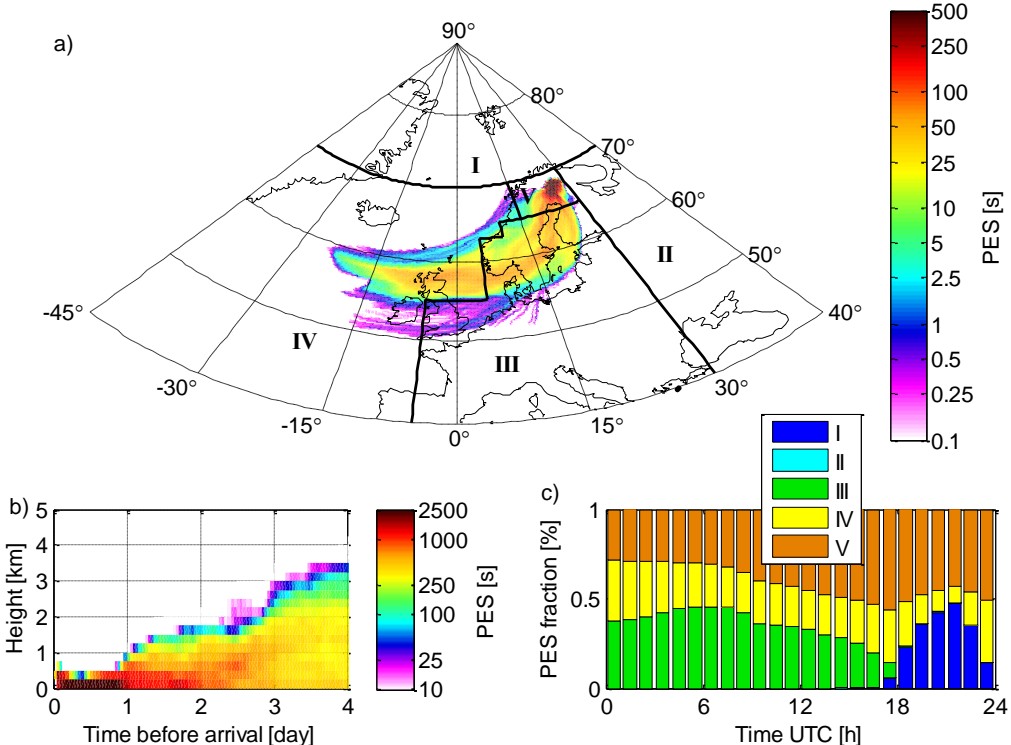

Figure 4: a) Horizontal distribution of vertically integrated PES for air masses arriving at Sammaltunturi on 20.09.2012 11:00 UTC. Source regions are indicated with Roman numerals. b) Vertical distribution
of PES in panel a as function of time before arrival at Sammaltunturi. c) Timeline of the PES fraction for each source region for 20.09.2012.

### 3.Results

3.1 Local meteorological conditions

Figure 5 shows the daily average temperatures at 570 m observed at the Sammaltunturi measuring site for days with "cloud events". Days with "cloud events" were defined as the days when the station was at least 30 minutes immersed in a cloud. To identify the presence of a cloud at the station, four steps were followed. The droplet size distribution was checked from both cloud spectrometers; the relative humidity should be ~ 100 %; the horizontal visibility should be less than 1000 m, and a final inspection
was performed visually using pictures recorded by an automatic weather camera installed on the roof of the station. During days with no cloud events, clouds could still exist at higher altitudes. Supercooled

water droplets were expected at temperatures <0 °C (usually during November and October of each campaign, in total 175 cloud events with temperature <0 °C were sampled). Mixed-phase clouds, consisting of water vapor, ice particles and supercooled liquid droplets, are frequent at temperatures

from −10 to −25 °C (Korolev et al., 2017, Filioglou et al., 2019); however, they can be present up to temperatures of 0°C (Andronache, 2017). During September, the average temperature was >0 °C, thus, the clouds were expected to consist of liquid hydrometeors only (liquid droplets, drizzle drops and raindrops). Wind speed ranges during the PaCEs were approximately from 0 to 10 m s$^{-1}$ and the average wind speed during each campaign was around 7 m s$^{-1}$. These values were lower in comparison to the

probe air speed of both cloud ground-based spectrometers (Doulgeris et al., 2022).

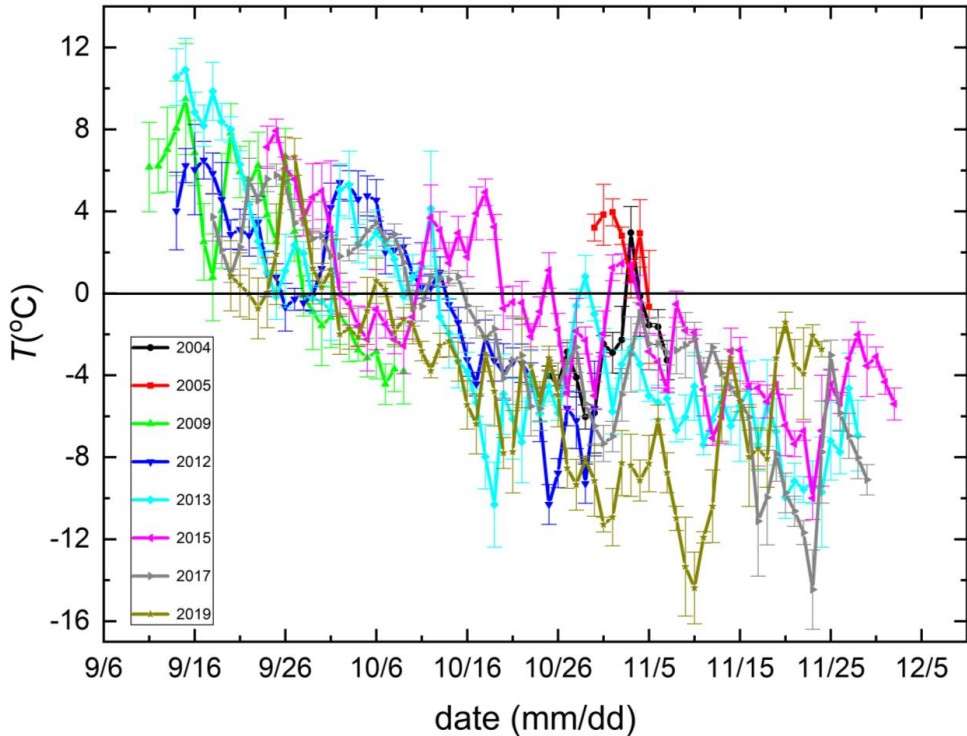

Figure 5: The daily averaged temperatures at the Sammaltunturi site for days with cloud events. The black solid line is used as a reference line for 0 °C temperature. The definition of a cloud event is provided in the text.


### 3.2. Identification of the air mass origin and its effect on the number concentration of cloud droplets.

First, we examined which was the optimal threshold of the PES fraction within one region that should be used for this region to be representative of the air mass type. $N_c$ was chosen to be used as a benchmark

parameter. The results are shown in Fig. 6. Figure 6a shows hourly $N_c$ values depending on the PES



fraction of the Arctic region (marine environment with no anthropogenic emissions). The lowest values of $N_c$ (< 30 cm$^{-3}$) were related to PES fractions > 80%. When the PES fraction was between 70 and 80%, the values of $N_c$ were varying between 5 and 80 cm$^{-3}$ and the highest values of $N_c$ values (> 30 cm$^{-3}$) were related to PES fraction lower than 70%.

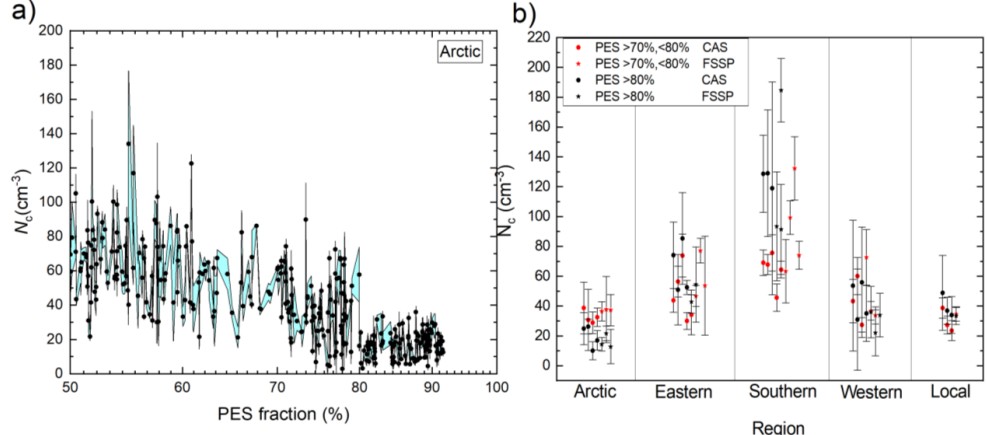


Figure 6: a) Hourly cloud droplet number concentration ($N_c$) versus the different potential emission sensitivity (PES) fraction for the Arctic region and b) $N_c$ for each region and single PaCE campaign as they were measured by the cloud and aerosol spectrometer (CAS) and the forward-scattering spectrometer probe (FSSP) where the PES fraction was within one region >80 % (black) and the PES fraction was within one region from 70 to 80 % (red).


To achieve a generalization of the large-scale air mass influence on microphysical cloud properties, the characteristics of all regions were intercompared. In Fig. 6b we summarize cloud $N_c$ measurements from both CAPS and FSSP. Each point represents a single PaCE campaign for different regions according to the classification criteria that were introduced previously in this section. Two different levels of the PES fraction were chosen to be further investigated; i) when the PES fraction was within one region more than 80% and ii) when the PES fraction was within one region between 70 and 80 %. For the first level, the highest values of $N_c$ (approximately 100-200 cm$^{-3}$) were clearly associated with Southern air masses, whereas the lowest ones (approximately 20 cm$^{-3}$) were observed in Arctic air masses. In general, marine air masses (Arctic, Western) arriving at Sammaltunturi resulted in lower values of $N_c$ compared with continental air masses. However, there was also a difference (approximately by a factor of 2) between marine air masses: $N_c$ in air masses travelling over the Atlantic Ocean, or the Norwegian Sea, were higher than those in air masses arriving from the Arctic Sea. For aerosol populations, higher values of $N_{CCN}$ are expected in more polluted air masses (Southern and Eastern air masses, due to emissions from Europe and Kola peninsula) (Virkula et al.,1997; Jaatinen et al., 2014; Sipila et al., 2021). In this work, $N_{CCN}$ measurements at different supersaturations were not conducted.





For PES fraction between 70 and 80 %, the impact of the air mass type on the $N_c$ changed as the differences in $N_c$ were less than for PES > 80%. (Fig. 5b). Clouds that were associated with Southern air masses had slightly higher values of $N_c$ (approximately 60-80 cm$^{-3}$) in comparison with clouds from the other regions (approximately 20-60 cm$^{-3}$). As a result, for further analysis in this work, we decided to exclude measurements that were performed when the PES fraction was between 70 and 80 %. Thus, we considered that > 80 % of the PES fraction within a particular region would be the optimal threshold to represent an air mass type during PaCEs. Using the > 80% PES fraction from one source region as a criterion for further analysis left 492 hours of in-cloud measurements with the CAS and 214 hours of in-cloud measurements with the FSSP probe, respectively, which ensured statistically robust results. Cloud observation related to Arctic, Eastern, Southern, Western and Local air masses were 118, 275, 152, 118 and 43 hours, respectively. Clouds that were related with local air masses were excluded due to relatively small number of observations.

Based on the air mass origin classification, a statistical analysis was made to investigate the frequency of the air masses during cloud events at the measuring site. When the air masses were not mixed, the occurrence of clouds at the station related to continental and marine air masses in 31.9 % and 14.3 % of the cases, respectively. Focusing on each region separately, 29.6 % of the cloud's occurrence seemed to be related to Southern and Eastern air masses and 7.4 % were related to Arctic air masses, although the predominant air mass at Sammaltunturi was from the Arctic (Asmi et al., 2012).

**3.3. Effect of the air mass origin on the cloud droplets size.**

In this section, we focused on investigating the size distribution of the cloud droplets and the derived parameters ED, MVD and LWC. ED and MVD are strongly dependent on the shape of the cloud droplet size distribution, while LWC is a function of both $N_c$ and sizes of the cloud droplets. To achieve a generalization, size distributions of cloud droplets related to each air mass origin for each single PaCE campaign are presented in Figure 7. Cloud droplet size distributions originating from marine regions (Arctic, Western) had a relatively broad shape with presence of large (10–20 µm) droplets, whereas in continental air masses there was a clear absence of large cloud droplets. In general, the average size distribution showed a spectrum with more droplets at small size ranges when the masses were continental and more droplets in larger size ranges when the air masses were marine. Cloud droplets larger than 16.0 µm started to appear in clouds that were characterizes by Arctic and western air masses. On the other hand, clouds that were characterized by Eastern and Southern air masses had cloud droplets mainly in the range from 5 to 10 µm. Values of $N_c$ for different sizes of the cloud droplets suggest that higher aerosol loadings lead to higher number concentrations of cloud droplets and smaller cloud droplet effective diameters. This result is consistent with the Twomey effect (Twomey, 1977), reported in several *in situ* observations (e.g. Twohy et al., 2005; Freud et al., 2008; Goren and Rosenfeld 2014;





Paramonov et al., 2015). In general, in a cloud system, it is expected that for a relatively constant LWC, the effective diameter of cloud droplets decreases as their number concentration increases.


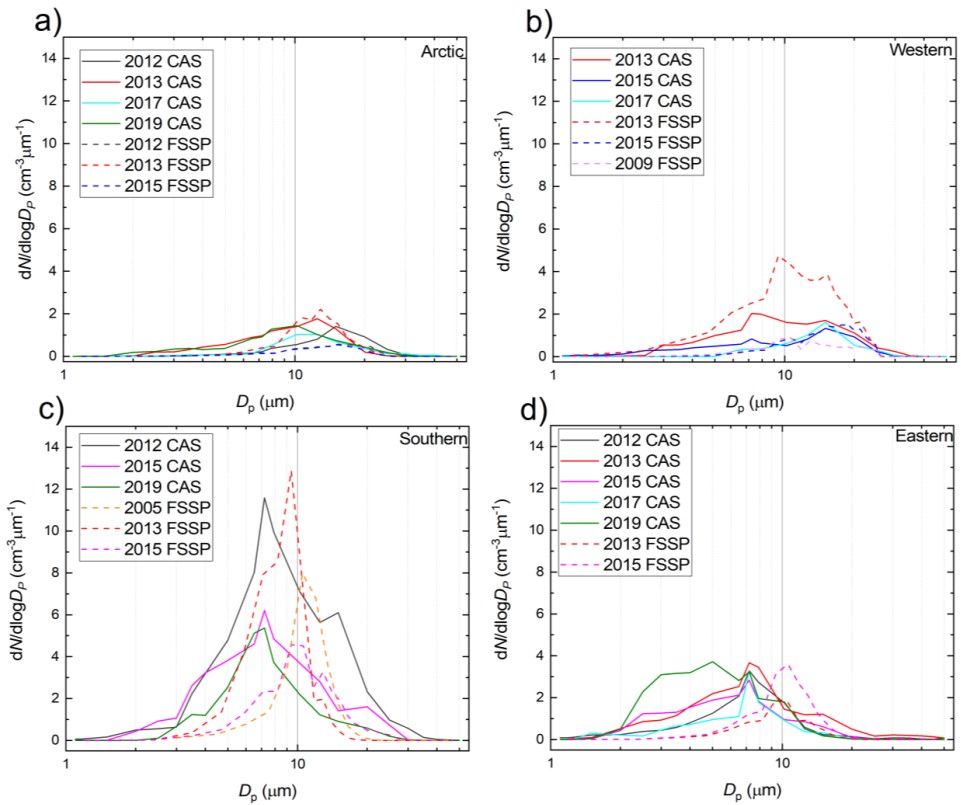

Figure 7. Yearly average cloud droplet size distribution associated with the (a) Arctic, (b) Western, (c) Southern and (d) Eastern region as they were measured by the cloud and aerosol spectrometer (CAS) and the forward-scattering spectrometer probe (FSSP).


We also investigated whether sizes of cloud droplets depend on the air temperature. For that reason, temperature bins of 4 °C range were created. Thus, the measurements were grouped at temperature bins of –10 to –6 °C, –6 to –2 °C, –2 to 2 °C and 2 to 6 °C. The mid temperature value of each bin was used to create Fig. 8 which shows that cloud droplets appeared to be more prone to grow

at temperatures larger than –2 °C. A hypothesis to explain such growth could be the collision–coalescence procedures that can take place in warm clouds (e.g., Xue et al., 2008; Pruppacher and Klett 2010; Lohmann et al., 2016). In this study, all the sampled clouds are considered as warm clouds, however those at warmer air temperatures seem to consist of larger droplets. Both MVD and ED had a similar behavior. When the clouds were characterized by Arctic air masses, MVD and ED were





approximately 15 µm within our temperature spectrum. This could possibly be related to the fact that warm clouds that were associated with Arctic air masses had smaller lifetimes compared to the other regions within our dataset. Clouds related to eastern air masses showed approximately 5 µm larger hydrometeors in warm clouds. However, more observations on existing and wider temperature ranges are needed to statistically ensure those results. In our case, we obtained approximately from 11 to 99

hours of observations for each temperature bin.

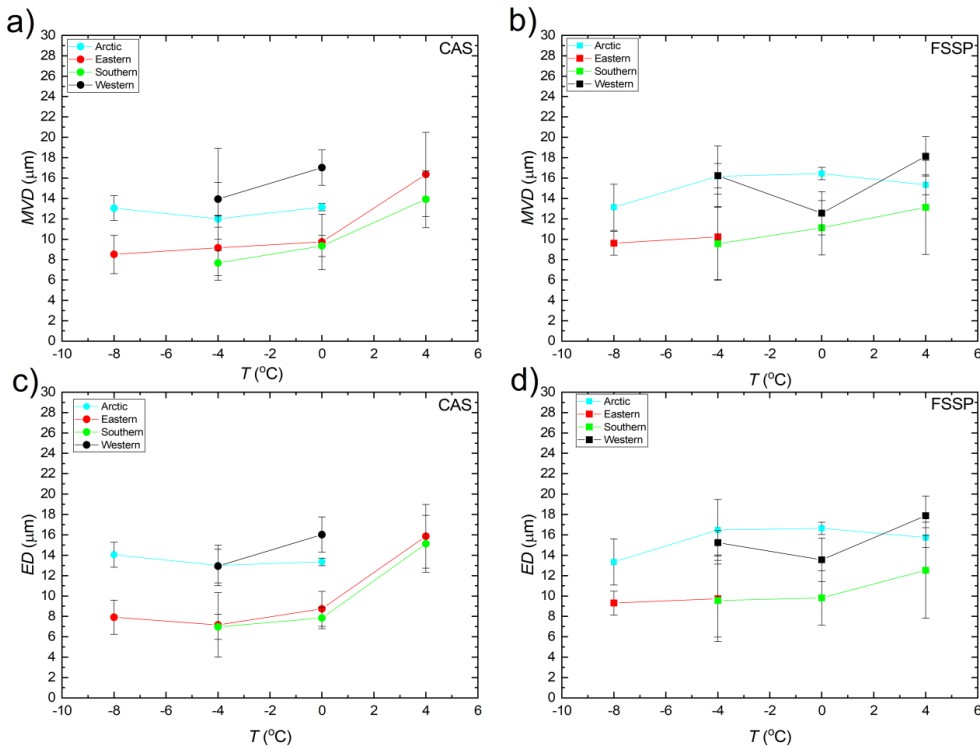

Figure 8. Hourly averages of median volume diameter (MVD) and effective diameter (ED) values versus temperature for all PaCE campaigns as they were measured by the cloud and aerosol

spectrometer (CAS) and the forward-scattering spectrometer probe (FSSP) where the PES fraction was within one region >80 %. Solid lines were made to lead readers' eyes.

LWC of low-level clouds for the different air mass types are summarized for each PaCE campaign (Fig.9a). The Arctic air masses were related to the lowest values of LWC (approximately

0.025 g m$^{-3}$), whereas the Southern air masses were related to the highest values of LWC (> 0.05g m$^{-3}$). Western and Eastern air masses were related to LWC values of approximately 0.025 to 0.05 g m$^{-3}$. In this study, LWC of continental air masses were, on average, larger than those of marine air masses.

This is reflected in the higher $N_c$ of continental air masses (Fig. 6b), as LWC is a function of both $N_c$ and size of cloud droplets. In Figure 9b, the relation between the $N_c$ and MVD is plotted. The points

were divided into three different levels according to the measured LWC. The values of MVD were ranging from ~ 9 to 19 µm. MVD was larger for higher values of LWC and decreased with an increasing cloud droplet number concentration for each LWC category. The LWC values of the clouds we sampled (~0.03 g m$^{-3}$ for marine and ~0.06 g m$^{-3}$ for continental conditions) are comparable to those observed in several other *in situ* cloud studies (e.g., Gultepe and Isaac 1997; Zhao et al., 2012; Lu et al.,2014;

Guyot et al., 2015; Dione et., al 2020).

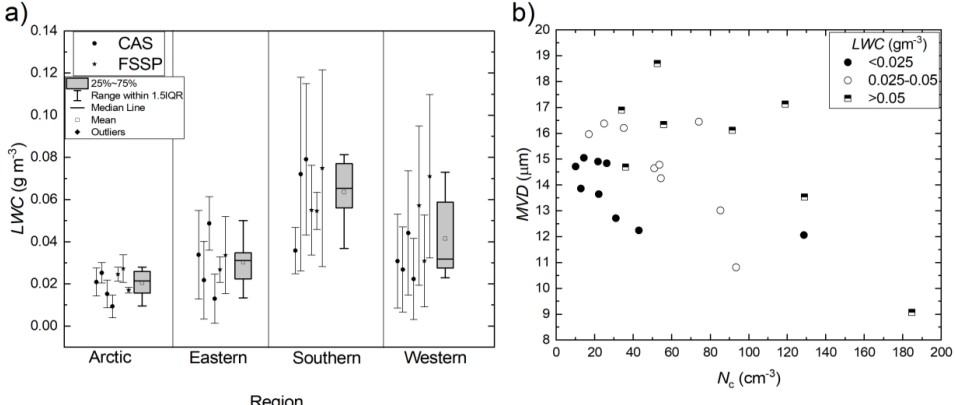

Figure 9: a) Liquid water content (LWC) as they were measured by the cloud and aerosol spectrometer (CAS) and the forward-scattering spectrometer probe (FSSP) where PES was within one region >80 %. The box plot for each region is also provided, b) Median volume diameter (MVD) as a function of total

cloud droplets number concentration ($N_c$) for three different categories of LWC. Each point represents a single PaCE campaign for different regions.

### 3.4 Influence of the vertical position of probe on the derived parameters

In this section, we focus on investigating how the derived parameters change with changes in the vertical position (altitude) of the sampling probe. Naturally, differences in the cloud base could affect the microphysical properties of a cloud, as seen in several studies (e.g., Martins et al., 2011; McDonald et al., 2018; Alexandrov et al., 2020). Under theoretical adiabatic conditions, the vertical profile of LWC is expected to increase linearly with height above cloud base, with a constant gradient that is dependent

on the temperature and pressure at cloud base (Brenguier, 1991). $N_c$ is constant through the vertical profile of the cloud layer, while the size of the droplets increases with altitude. Assuming homogenous mixing, this expectation of the cloud microphysical profile also holds for 'scaled-adiabatic' conditions which include the entrainment of drier air (Boers et al., 2000). In reality, there are more processes to consider, which lead to departures from this ideal condition, particularly towards cloud top (Pawlowska

et a2l., 2006). As already discussed in Section 2.2., both ground-based spectrometers were fixed in one vertical position. Thus, there were cases that we sampled different layers of a cloud in a range of 120 meters from the cloud base. The ground-based spectrometers were placed at the Sammaltunturi, 210 m above the ceilometer (installed at the Kenttärova site). Kenttärova is located 4.3 km to the east of hilltop Sammaltunturi station. Since the Sammaltunturi station is on a top of an Arctic fell, cloud formation

and properties could also be influenced by the local topography via changes in turbulence or orographic flows. The ceilometer's resolution in estimation of the cloud base was 30 meters.

    In Fig. 10, both MVD and ED are presented as they were derived from both cloud spectrometers for different altitudes of the cloud base. The distance of the cloud spectrometer was relative to the cloud base. From this analysis, it is apparent that there was no strong dependency between the vertical position

of CAS and FSSP in the cloud and the derived sizing parameters. However, it is expected that number concentration provides a robust signal, whereas MVD and ED has some extra uncertainties depending on the altitude with respect to cloud base.

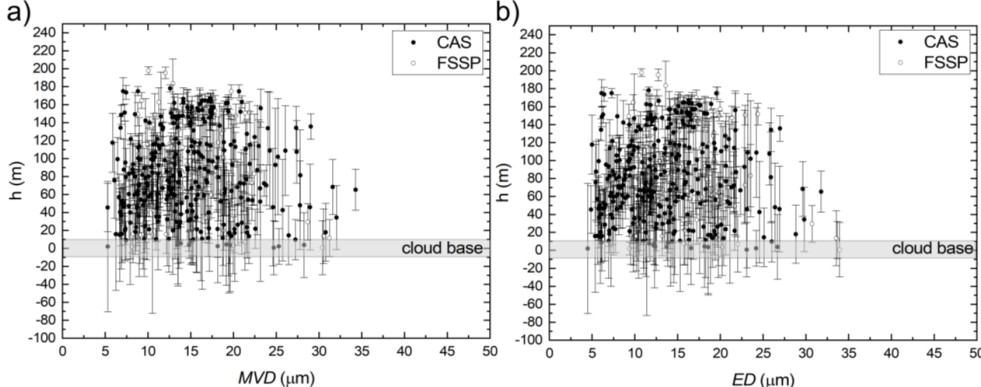

Figure 10. The cloud base height (h) (relative distance of the cloud ground-based spectrometer) as it

was measured at the Kenttärova station versus hourly averages of median volume diameter (MVD) and effective diameter (ED) values for all PaCEs as they were measured by the cloud and aerosol spectrometer (CAS) and the forward-scattering spectrometer probe (FSSP) where PES was within one region >80 %.

**4. Summary and conclusions**

Our main goal during this work was to quantify the effect of air mass origin on cloud microphysical properties in a clean subarctic environment. Thus, the impact of different air masses on cloud properties in the subarctic Finland was investigated based on data from 8 Pallas Cloud Experiments (PaCEs) made

during 2004–2019. For measuring the cloud microphysical properties, we deployed two cloud ground-based spectrometer probes: the cloud and aerosol spectrometer and the forward-scattering spectrometer



probe. For performing the air mass sources classification, the FLEXPART model was used with ERA5 meteorology. The air mass source regions were categorized into Arctic, Eastern, Southern, Western and Local sectors, with the Arctic and Western sectors representing marine air masses, and the Eastern,

Southern and Local sectors representing continental airmasses.

Our analysis demonstrated that different air mass types had significant impacts on cloud microphysics. When 80 % of the potential emission sensitivity fraction was within a region, the observations were considered to representative of that air mass type. The occurrence of a cloud at the station was more frequent with air masses arriving from the Southern and Eastern regions than other

source regions. Furthermore, continental air masses led to the highest cloud droplet number concentrations (~ 100-200 cm$^{-3}$) and marine air masses to the lowest ones (~ 20 cm$^{-3}$). Number concentration was expected to be a robust signal with no dependency from the vertical position of cloud spectrometer in the cloud. On the other hand, both effective radius and median volume diameter has some extra uncertainties depending on the altitude with respect to cloud base. In general, the median

volume diameter and effective radius of cloud droplets was found to be influenced by the cloud droplet number concentration: Clouds associated with marine air masses had larger cloud droplets (ranging from 15 to 20 µm) in comparison with continental clouds (ranging from 8 to 12 µm). These results are in agreement with the Twomey effect (Twomey, 1977). Furthermore, there was an indication that cloud droplets in clouds in warmer air (from –2 to 6 °C) were more prone to grow. However, more

measurements are needed to confirm such temperature dependency of droplet sizes.

**Data availability**

The cloud probes and meteorological data used here are available in the Finnish Meteorological Institute (FMI) open data repository for each campaign and each cloud spectrometer ground setup individually

(Doulgeris et al., 2021; Doulgeris et al., 2022). The FLEXPART simulations and the ceilometer dataset are available upon request to the corresponding author (konstantinos.doulgeris@fmi.fi).

*Author contributions.* KD wrote the paper with contributions from all co-authors. HL planned and coordinated PaCE 2004, 2005, and 2009. HL and DB planned and coordinated PaCE 2012 and 2013.

KMD and DB planned and coordinated PaCE 2015, 2017, and 2019. KMD and DB processed, analyzed, and quality controlled the data set. VV carried out the FLEXPART simulations. EOC provided the ceilometer data. VMK reviewed and edited the manuscript.

*Competing interests.* The authors declare no conflict of interest.


*Acknowledgements:* This work was supported by the Koneen Säätiö(grant no. 46-6817), Academy of Finland Flagship funding (grant no. 337552). This project has also received funding from the European Union, H2020 research and innovation program (ACTRIS-IMP, the European Research Infrastructure



for the observation of Aerosol, Clouds, and Trace gases, no. 871115). The authors wish to acknowledge
CSC – IT Center for Science, Finland, for computational resources. The authors would also like to thank
all the people who help in PACE campaign measurements throughout the years.

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

Shupe, MD, Rex, M, Blomquist, B, Persson, POG, Schmale, J, Uttal,T, Althausen, D, Angot, H, Archer, S,
Bariteau, L, Beck, I, Bilberry, J, Bucci, S, Buck, C, Boyer, M, Brasseur, Z, Brooks, IM, Calmer, R, Cassano, J,
Castro, V, Chu, D, Costa, D, Cox, CJ, Creamean, J, Crewell, S, Dahlke, S, Damm, E, de Boer, G, Deckelmann,
H, Dethloff, K, Du¨tsch, M, Ebell, K, Ehrlich, A, Ellis, J, Engelmann, R, Fong, AA, Frey, MM, Gallagher, MR,
Ganzeveld, L, Gradinger, R, Graeser, J, Greenamyer, V, Griesche, H, Griffiths, S, Hamilton, J, Heinemann, G,
Helmig, D, Herber, A, Heuze´, C, Hofer, J, Houchens, T, Howard, D, Inoue, J, Jacobi, H-W, Jaiser, R, Jokinen,
T, Jourdan, O, Jozef, G, King, W, Kirchgaessner, A, Klingebiel, M, Krassovski, M, Krumpen, T, Lampert, A,
Landing, W, Laurila, T, Lawrence, D, Lonardi, M, Loose, B, Lu¨pkes, C, Maahn, M, Macke, A, Maslowski, W,
Marsay, C, Maturilli, M, Mech, M, Morris, S, Moser, M, Nicolaus, M, Ortega, P, Osborn, J, Pa¨tzold, F,
Perovich, DK, Peta¨ja¨,T, Pilz, C, Pirazzini, R, Posman, K, Powers, H, Pratt, KA, Preußer, A, Que´le´ ver, L,
Radenz, M, Rabe, B, Rinke, A, Sachs, T, Schulz, A, Siebert, H, Silva, T, Solomon, A, Sommerfeld, A, Spreen,
G, Stephens, M, Stohl, A, Svensson, G, Uin, J,Viegas, J,Voigt, C, von der Gathen, P, Wehner, B, Welker, JM,
Wendisch, M, Werner, M, Xie, ZQ, Yue, F. 2022. Overview of the MOSAiC expedition: Atmosphere.
Elementa: Science of the Anthropocene 10(1). DOI: https://doi.org/10.1525/elementa.2021.00060

Sipilä, M., Sarnela, N., Neitola, K., Laitinen, T., Kemppainen, D., Beck, L., Duplissy, E.-M., Kuittinen, S.,
Lehmusjärvi, T., Lampilahti, J., Kerminen, V.-M., Lehtipalo, K., Aalto, P. P., Keronen, P., Siivola, E., Rantala,
P. A., Worsnop, D. R., Kulmala, M., Jokinen, T., and Petäjä, T.: Wintertime subarctic new particle formation
from Kola Peninsula sulfur emissions, Atmos. Chem. Phys., 21, 17559–17576, https://doi.org/10.5194/acp-21-
17559-2021, 2021.

Small, J. D., Chuang, P. Y., Feingold, G., and Jiang, H.: Can aerosol decrease cloud lifetime?, Geophys. Res.
Lett., 36, L16806, https://doi.org/10.1029/2009GL038888, 2009.

Solomon, A., & Shupe, M. D. (2019). A Case Study of Airmass Transformation and Cloud Formation at
Summit, Greenland, Journal of the Atmospheric Sciences, 76(10), 3095-3113. Retrieved May 9, 2022,
from https://journals.ametsoc.org/view/journals/atsc/76/10/jas-d-19-0056.1.xml

Stohl, A., Forster, C., Frank, A., Seibert, P., and Wotawa, G.: Technical note: The Lagrangian particle
dispersion model FLEXPART version 6.2, Atmos. Chem. Phys., 5, 2461–2474, https://doi.org/10.5194/acp-5-
2461-2005, 2005.

Torres-Delgado, E., Baumgardner, D., and Mayol-Bracero, O. L.: Measurement report: Impact of African
aerosol particles on cloud evolution in a tropical montane cloud forest in the Caribbean, Atmos. Chem. Phys.,
21, 18011–18027, https://doi.org/10.5194/acp-21-18011-2021, 2021.

Twohy, C. H., M. D. Petters, J. R. Snider, B. Stevens, W. Tahnk, M. Wetzel, L. Russell, and F. Burnet (2005),
Evaluation of the aerosol indirect effect in marine stratocumulus clouds: Droplet number, size, liquid water path,
and radiative impact, J. Geophys. Res., 110, D08203, doi:10.1029/2004JD005116.

Vaisala Oyj, Ceilometer CT25K: User's Guide; Vaisala Oyj: Vantaa, Finland, 2002.

Virkkula, A., Hillamo, R. E., Kerminen, V.-M. & Stohl, A. 1997. The influence of Kola Peninsula, continental
European and marine sources on the number concentrations and scattering coefficients of the atmospheric aerosol
in Finnish Lapland. Boreal Env. Res. 2: 317–336. ISSN 1239-6095.

Wandinger, U., Apituley, A., Blumenstock, T., Bukowiecki, N., Cammas, J.-P., Connolly, P., De Mazière, M.,
Dils, B., Fiebig, M., Freney, E., Gallagher, M., Godin-Beekmann, S., Goloub, P., Gysel, M., Haeffelin, M.,
Hase, F., Hermann, M., Herrmann, H., Jokinen, T., Komppula, M., Kubistin, D., Langerock, B., Lihavainen, H.,
Mihalopoulos, N., Laj, P., Lund Myhre, C., Mahieu, E., Mertes, S., Möhler, O., Mona, L., Nicolae, D.,
O'Connor, E., Palm, M., Pappalardo, G., Pazmino, A., Petäjä, T., Philippin, S., Plass-Duelmer, C., Pospichal, B.,
Putaud, J.-P., Reimann, S., Rohrer, F., Russchenberg, H., Sauvage, S., Sellegri, K., Steinbrecher, R., Stratmann,
F., Sussmann, R., Van Pinxteren, D., Van Roozendael M., Vigouroux C., Walden C., Wegene R., and
Wiedensohler, A.: ACTRIS-PPP Deliverable D5.1: Documentation on technical concepts and requirements for



ACTRIS Observational Platforms, available at: https://www.actris.eu/Portals/46/Documentation/ACTRIS PPP/Deliverables/Public/WP5_D5.1_M18.pdf?ver= 2018-06-28-125343-273 (last access: 21 april 2022), 2018.

Wang, Y., Zheng, X., Dong, X., Xi, B., Wu, P., Logan, T., and Yung, Y. L.: Impacts of long-range transport of aerosols on marine-boundary-layer clouds in the eastern North Atlantic, Atmos. Chem. Phys., 20, 14741–14755, https://doi.org/10.5194/acp-20-14741-2020, 2020.

Wendisch, M., Macke, A., Ehrlich, A., Lüpkes, C., Mech, M., Chechin, D., Dethloff, K., Velasco, C. B., Bozem, H., Brückner, M., Clemen, H.-C., Crewell, S., Donth, T., Dupuy, R., Ebell, K., Egerer, U., Engelmann, R., Engler, C., Eppers, O., Gehrmann, M., Gong, X., Gottschalk, M., Gourbeyre, C., Griesche, H., Hartmann, J., Hartmann, M., Heinold, B., Herber, A., Herrmann, H., Heygster, G., Hoor, P., Jafariserajehlou, S., Jäkel, E., Järvinen, E., Jourdan, O., Kästner, U., Kecorius, S., Knudsen, E. M., Köllner, F., Kretzschmar, J., Lelli, L., Leroy, D., Maturilli, M., Mei, L., Mertes, S., Mioche, G., Neuber, R., Nicolaus, M., Nomokonova, T., Notholt, J., Palm, M., van Pinxteren, M., Quaas, J., Richter, P., Ruiz-Donoso, E., Schäfer, M., Schmieder, K., Schnaiter, M., Schneider, J., Schwarzenböck, A., Seifert, P., Shupe, M. D., Siebert, H., Spreen, G., Stapf, J., Stratmann, F., Vogl, T., Welti, A., Wex, H., Wiedensohler, A., Zanatta, M., and Zeppenfeld, S.: The Arctic Cloud Puzzle: Using ACLOUD/PASCAL Multiplatform Observations to Unravel the Role of Clouds and Aerosol Particles in Arctic Amplification, B. Am. Meteorol. Soc., 100, 841–871, https://doi.org/10.1175/BAMS-D-18-0072.1, 2019.

Jaatinen, A., Romakkaniemi, S., Anttila, T, Hyvärinen, A.-P., Hao, L. Q., Kortelainen, A., Miettinen, P., Mikkonen, S., Smith, J. N., Virtanen, A. & Laaksonen, A. 2014: The third Pallas Cloud Experiment: Consistency between the aerosol hygroscopic growth and CCN activity. Boreal Env. Res. 19 (suppl. B): 368–382.

Xue, Y., Wang, L., and Grabowski, W. W. : Growth of Cloud Droplets by Turbulent Collision–Coalescence, Journal of the Atmospheric Sciences, 65(2), 331-356. https://doi.org/10.1175/2007JAS2406.1, 2008.

Zhao, C.F., Xie, S. C., Klein, S.A., Protat, A., Shupe, M.D., McFarlane, S.A,Comstock, J.M., Delanoe, J., Deng, M., Dunn, M., Hogan, R.J., Huang, D., Jensen, M.P., Mace, G.G., McCoy, R., O'Connor, E.J., Turner, D.D. and Wang, Z.E.: Toward understanding of differences in current cloud retrievals of ARM ground-based measurements. J. Geophys. Res.-Atmos., 117, p. D10206, 10.1029/2011jd016792. 2012.