# Peer review of "Influence of air mass origin on microphysical properties of lowlevel clouds in a subarctic environment."

_Atmospheric Chemistry and Physics, 2022_

## Author Comment (AC1)

We sincerely thank reviewer #1 for carefully reading our manuscript, and for their review and constructive comments. We have reviewed the comments and have revised the manuscript accordingly. Our response is given in a point-by-point manner below. Reviewer comment (RC) and authors answer (AA).

*In the study "Influence of air mass origin on microphysical properties of low-level clouds in a subarctic environment" by Doulgeris et al. microphysical cloud properties measured during eight Pallas Cloud Experiments in the Finnish subarctic region are analyzed with respect to their air mass origin based on the Lagrangian particle dispersion model FLEXPART.*

*The scientific approach is valid and the manuscript is structured in a clear and concise way.*

*However, two main deficits regarding the scientific relevance and thus the scientific quality of the study as described in the general comments would require major revisions.*

**General comments**

RC1: "*1. Scientific relevance. The study is based on a large time series of measurements campaigns that have been conducted in a subarctic mostly pristine region adequate for the analysis of aerosol cloud-interactions (ACI). A clear statement is missing on how the presented results may advance the current state of the art. The dependence of cloud microphysics on the air mass origin (Twomey effect in continental air masses vs. marine air masses) and that cloud droplets are prone to grow in warmer air is known already from other studies (cited in the manuscript l.346). Also, the introduction is not clearly leading to a research hypothesis or research question. Modifications in the Introduction and discussion of results as well as in the abstract and conclusions are required to specify the scientific relevance of the study in the context of existing literature. The identification and further interpretation of results that add new findings to the existing body of knowledge would be helpful for the ACI community and further studies. *"

AA1: We have modified the text to be clearer with the main goals and significance of this work. Cloud processes are considered an important component of climate change in the Arctic region (Wendisch et al., 2019). However, even though there is an increased demand for long term continuous ground based in-situ cloud measurements, unfortunately there is limited instrumentation to cover such demand. The atmospheric in-situ measurements community (in our case the European Research Infrastructure for the observation of Aerosol, Clouds and Trace Gases, ACTRIS) has identified the cloud droplet probes with surface installation as a potential method for continuous cloud in situ measurements (ACTRIS-PPP Deliverable D5.1: Documentation on technical concepts and requirements for ACTRIS Observational Platforms). However, measurements in conditions like those at our sub-Arctic location are very challenging. As a result, the dependence of cloud microphysics on the air mass origin in a subarctic mostly pristine region were rarely seen until now using such an in situ long term dataset. We agree with the reviewer that there were already excellent studies that investigates cloud microphysics and their connection to air mass origin (e.g., Fuchs et al. 2017; Cho et al., 2021). However, one of the main differences with those studies is the methodology used. E.g., Fuchs et al. (2017) explain the importance of air mass origin and its characteristics to cloud properties using satellite data. Cho et al. (2021) investigate the relationship of cloud properties and radiative effects with air mass origin during the

winter (from November to February, 2016–2020) at Ny-Ålesund, Svalbard, by means of s remote sensing approach using a combination of cloud radar, ceilometer, and microwave radiometer measurements. To our knowledge, this is the first study that connects extensive in situ cloud measurements to air mass origin. In this work, we point out that there is need to consider not only local meteorological parameters but also the air mass origin in investigations of cloud processes. The PaCE measuring period (during autumn) is crucial as it is a unique opportunity for both experiencing Arctic pristine air masses (Pernov et al., 2022) and being able to measure them in-situ with ground-based cloud instrumentation. Moreover, the procedure of distinguishing cases that corresponds to a single air mass origin and not a mixed one is complicated and require a huge amount of continuous data. In this work, in situ cloud data with ground-based cloud spectrometers from eight different autumn campaigns were obtained (2004 hours of cloud observations resulted in 706 hours of cloud observations that related to all clean air mass origins). From this dataset, the relationship between $N_c$ and droplet size (i.e., the Twomey effect) was characterized for the different source regions. We proved that cloud microphysical properties and particularly the number concentration of cloud droplets have a strong dependence on the air mass origin. Using those findings, the ACI community can focus on further studies to investigate how aerosol and meteorology of different airmasses along with local meteorological parameters change the cloud microphysics and to what scale.

Some of the major changes were applied to abstract, line 22

"…Local). We observed clear differences in the cloud microphysical properties for the air mass source regions. Arctic air masses were characterized by low liquid water content (LWC), low cloud droplet number concentration ($N_c$), and comparatively large median volume and effective droplet diameter. Western region (marine North Atlantic) differed from Arctic by both higher $N_c$ and LWC. Eastern region (continental Eurasia) had only a little higher LWC than Arctic, but substantially higher Nc and smaller droplet diameter. Southern region (continental Europe) had high $N_c$ and LWC, and very similar droplet diameter to the Eastern region. Finally, the relationship between $N_c$ and droplet size (i.e., the Twomey effect) was characterized for the different source regions, indicating that all region clouds were sensitive to increases in $N_c$."

To introduction, line 92 "…at Pallas. To our knowledge, this is the first study that connects extensive in situ cloud measurements to air mass origin. During autumn, clean, natural Arctic background conditions are significantly increasing (Pernov et al., 2022). Subsequently, this allows us to focus in this work on quantifying the impact of air mass origin (e.g., clean arctic vs. long-range transported air from continental Europe) on the microphysical properties of low-level clouds and their patterns based on measurements at the Pallas GAW station. To our knowledge, this is the first study that connects extensive in situ cloud measurements to air mass origin."

To results, line 285 "(a marine environment that the natural Arctic background conditions are significantly increasing (Pernov et al., 2022)"

Line 321 "Averaged temperatures at Sammaltunturi for each air mass were -3.1°C (SD 2.5°C), -2.2 °C (SD 5.9 °C), 1.3 °C (SD 3.9 °C) and -2.8 °C (SD 2.01 °C) for the arctic, eastern, southern and western region respectively. Furthermore, in all regions, there was no clear indication that there was any trend in $N_c$ through different years of PaCEs."

To summary and conclusions line 475, 478 and 488

"This result suggests that clouds occurrence depended on the different meteorological conditions that were associated with the different air parcels"

"The lowest values of cloud droplet concentration were related to clean arctic airmasses. According to theoretical considerations (Brenguier 1991; Pawlowska et al., 2006), the measurements of cloud droplet number concentration do not depend on the vertical position of the cloud spectrometer within the cloud layer.

"The above differences that were observed in cloud microphysical properties when the air masses were related to different regions show the need to investigate how the aerosol loading and meteorology of different airmasses along with local meteorological parameters change the cloud microphysics and in which scale."

RC2:*" 2. Scientific approach. Cloud properties are analyzed according to their air mass origin in 5 predefined source regions. Cloud properties strongly depend on the air mass characteristics including humidity, wind speed, temperature etc. at different altitudes. Including air mass characteristics (e.g. from ERA5 reanalysis) in the analysis to understand differences in Nc, MVD, ED as started in Fig. 8 would make results more interpretable and scientifically relevant.*

*Also the approach of using predefined source regions is questioned as this classification may result in similar/mixed air mass characteristics as shown for the Eastern/Southern and Arctic/Western air masses in Fig. 8. More intuitive would be an automatic classification (grouping) based on the air-mass origins or pathways. "*

AA2:

Investigating cloud microphysical properties and revealing the main factors that are dependent and at what scale is a complex procedure and this is also highlighted in summary and conclusion section of the revised manuscript. In the revised manuscript, we include also meteorological information from each region as they were measured at the station. Our main scope in this work is to investigate low level clouds using in situ measurements and the influence of different air mass origin. We proved that the air mass origin significantly affects the number concentration of the cloud droplets (the aerosol loading from each region is expected to play a role in this case). When investigating the sizes of the cloud droplets, the sizes were influenced by the number concentration of the cloud droplets as suggested by the Twomey effect.

Although MVD is quite similar for the source region combinations as pointed out by the reviewer, the different source regions do stand out from each other when Nc and LWC are included in the comparison. The following figure and discussion were added in the supplementary material (SM) of the revised manuscript.

[Figure]

Figure S1. ERA5 temperature (T), specific humidity (Q) and wind speed (WS) profiles for the cases, when at least 80% of PES was within a source region. Line is the median and error bars indicate upper and lower quartiles. I, II, III and IV corresponds to the arctic, eastern, southern, and western region respectively. Station pressure is ~970 hPa.

Temperature (T), specific humidity (Q) and wind speed (WS) profiles from ERA5 for the different source regions (Fig. 1) were compared. In ERA5 profiles, Southern source region stands out as the one with higher T and Q, which is also reflected in the observed cloud microphysical properties. For Western and Eastern region, the median profiles are quite similar to the Arctic profile, but the interquartile range is wider. For these source regions we observe higher variability in e.g., LWC compared to the Arctic source region, which suggests more variable meteorological conditions for these source regions. For WS the differences are relatively small.

We agree with the reviewer that finding the most important source areas is usually a difficult task in a variable environment and should be done with prudence. In the revised manuscript, we will elaborate on our decision to use the predefined regions to make clear to the reader the methodology that was used. The division of the areas is predefined as it was based on previous studies that were conducted at Sammaltunturi (e.g., Aalto et.al., 2003, Asmi et al., 2011). This work is a continuation of those studies and for this reason we decided to adopt the same source areas. Initially, the regions were classified using trajectories cluster analysis, following the method as Eneroth et al. (2003) proposed. The predefined regions were used for different studies and scopes as atmospheric transport of carbon dioxide (Aalto et.al., 2003, Eneroth et al., 2005), aerosol studies (Tunved et al., 2006; Asmi et al., 2011). The choice of sectors represents roughly the characteristics of the region: the West and North are marine sectors, while the East and South are more continental sectors.

Line 224 "…Fig.3. The division was based on previous studies that were conducted at Sammaltunturi (e.g., Aalto et.al., 2003; Eneroth et al., 2005; Tunved et al.,2005, Asmi et al.,2011). Initially, the regions were classified using trajectories cluster analysis, following the method that Eneroth et al. (2003) proposed. The choice of sectors represents roughly the characteristics of the region. The Arctic… "

**Specific comments**

RC3: *"l. 106: Do you have information on the cloud type, is this mainly fog or low stratus? This may imply different processes."*

AA3: The cloud type is low stratus or stratocumulus. This is diagnosed from the ceilometer observations: first, liquid cloud should be present at both the mountain top station and in the ceilometer profile for a minimum specified duration (30 minutes); and secondly, the liquid cloud base should be above the ground at the ceilometer location (the altitude of which is 210 m below the mountain top station). This ensures that we are not observing fog (at the ceilometer location) and that, since the cloud layer has to be present over a larger area to be included in a cloud event (present at both locations at the same time for a minimum duration of 30 minutes) and not varying much in height with time, we can then assume that it is stratus/stratocumulus at both locations.

RC4: *"l. 123: Latitude and longitude is missing in Fig. 1."*

AA4: The above suggestion was accepted; A new map was created.

RC5: *"l. 201: Delete "model" as this can be mixed-up with numerical models.*

AA5: The above suggestion was accepted.

RC6: *"l. 207: it is not specified if the PES belongs to an aerosol type or Nc or which emission inventory is used to calculate the PES. If solely air mass trajectories are calculated backwards what is the PES referred to? Please provide more details on the FLEXPART model settings and assumptions here.*

AA6: Potential emission sensitivity (PES) is not connected to any emission inventory in this case, it is used only to characterize air mass history – therefore it is called "potential". While PES can be clustered to retrieve air mass trajectories similar to e.g., Hysplit, we have chosen to utilize the PES field directly in the source region classification, as this accounts for turbulent mixing during the transport (c.f. Fig. 4 in the manuscript). Within FLEXPART, PES is calculated from a retro-plume of inert tracers released at the measurement location and propagated backward in time. More information on PES can be found in Seibert and Frank (2004) and Pisso et al. (2019), which are referred to in the manuscript.

Lines 209-218 in the manuscript give already all the FLEXPART settings and describe the meteorological fields needed to re-run the simulations. Hence, we have made no changes to the manuscript.

"ERA5 reanalysis by European Centre for Medium-Range Weather Forecasts (ECMWF) was used as meteorological input fields for FLEXPART at 1 hour temporal resolution and 0.25° resolution in latitude and longitude. In vertical, ERA5 levels 50 to 137 were used, which corresponds approximately to the lowest 20 km above surface. The model domain was from 125° W to 75° E and 10° N to 85° N, which was large enough to contain 96 h simulations backward in time. FLEXPART runs were initiated at an hourly time resolution for the in-cloud measurement periods at Sammaltunturi. The retro plume release height was set to 560-660 m ASL, as the terrain height in ERA5 at the site was approximately 300 m ASL. The PES output resolution was set to 0.2° latitude and longitude with a 250-m height resolution up to 5 km and two additional output levels at 10 km and 50 km. "

RC7: "*l. 244: Subtitle 3.1 should be bold as 3.2 and 3*.3.

AA7: The above suggestion was accepted

RC8: "*l. 245: The main message of the figure is not mentioned and should include something like: It shows the seasonal range of temperatures from on average XX°C in September to -XX°C in November and its interannual variability.*

AA8: The above suggestion was accepted. The following text was added in the revised manuscript.

Line 253" ..events". The seasonal range of temperatures from on average 4.5 °C (SD 2.1°C) in September to -5.3 °C (SD 1.8°C) in November and its interannual variability is revealed. Days.."

RC9: *"Fig. 5 (also Fig. 7): Is there a reason to present each year separately? If not I suggest in accordance with the main message of the figure to present only one average line together with the standard deviation and include data gaps in the data section. This also applies to Fig. 7 and would increase clarity of the figures as 4 panels can even be summarized in one panel (4 lines - 4 regions). Data gaps and instrument specifications can be moved to the data section.*

AA9:

Both figures 5 and 7 were simplified as suggested by reviewer. One average line from all PaCEs was used in the figure of the revised manuscript. Thus, figure 5 was modified as

[Figure]

Manuscript figure 5: The daily averaged temperatures at the Sammaltunturi site for days with cloud events during all PaCE campaigns. The black solid line is used as a reference line for 0 °C temperature. The definition of a cloud event is provided in the text. The shaded area represents the corresponding standard deviations.

Combining the data would be ideal for understanding the size distribution that corresponds to each region. To simplify our results a new figure will be provided including the average size distribution from both instruments that corresponds to each region. The instruments are still presented separately since they have different bin sizes, thus they cannot be combined. Also, they represent different measurement periods as both spectrometers were not always working at the same time. Figure 7 (Figure 8 in the revised manuscript) was modified as

[Figure]

Figure 8. Cloud droplet size distribution associated with the (a) Arctic, (b) Western, (c) Southern and (d) Eastern region as they were measured by the cloud and aerosol spectrometer (CAS) and the forward-scattering spectrometer probe (FSSP) during all PaCEs. The shaded areas represent the corresponding standard deviations.

RC10: *" l. 271: anthropogenic aerosols: Is this an assumption, provide a reference?*

AA10:. Text was modified, and the following reference will be added.

Line 285 "... a marine environment that the natural Arctic background conditions are significantly increasing (Pernov et al., 2022))"

RC11: l. 275: *Fig. 6 a) What is the meaning of the cyan color? If not necessary please remove it. If it is representing a range, please indicate it in the legend.*

AA11: The shaded area represents the corresponding standard deviations. In the revised manuscript the use of shaded areas is explained.

RC12: l. 275: *Fig. 6 b) Symbols (stars and circles) representing different Nc measurements are difficult to distinguish. Would recommend either summarizing campaigns sorted by PES and*

*CAS/FSSP (4 symbols per air mass) or summarizing it even further only by PES. If this is no option, increasing maker size and distance between campaigns would improve clarity.*

AA12: The decision to present each PaCE was made due to each campaign had different operation times and the instruments could be also operative in different periods. We would like to keep each campaign and instrument to demonstrate that there were no obvious changes through years or possible malfunction of the instruments that were used and could produce biased results. However, the clarity of this figure should be improved. Thus, figure 6b was replaced and includes additional information to distinguish the years. A comment was added in the revised manuscript to note that there is no indication of dependence of the $N_c$ through different years of measurements.

[Figure]

Figure 7: Cloud droplet number concentration ($N_c$) for each region and single PaCE campaign as they were measured by the cloud and aerosol spectrometer (CAS) and the forward-scattering spectrometer probe (FSSP) where the PES fraction was within one region >80 % and the PES fraction was within one region from 70 to 80 %. Error bars indicate the corresponding standard deviation.

Line 308 "We present each campaign and instrument to demonstrate that there were no obvious changes through years or possible malfunction of the instruments that were used and could produce biased results".

Line 323 "...respectively. Furthermore, there was no clear indication that there was any trend in $N_c$ through different years of PaCEs."

RC13: *l. 284 CAPS --> CAS? (as in the legend of 6b), please check usage throughout the manuscript.*

AA13. In this work, only the CAS probe was used from the CAPS probe ground setup. Thus, CAS will be used in whole manuscript.

RC14: *l. 395-396 Fixed vertical position, but different layers? Something is missing: "...cases that we sampled WITH different layers".*

AA14: The typo was corrected, "with" was added.

RC15: l. *409 Fig 10 a and b. I cannot see the difference between CAS and FSSP in the plot. If the difference is not important for conveying the message that MVD/ED is not dependent on the position of the probe I would skip the legend entry. This figure could be improved by using a scatter density plot (2-D histogram) and regression line.*

*Further, If there is no dependence between MVD/ED and position of the probe, is this something still relevant for the main message of the paper and would it require a figure plus subsection? If the answer is no I recommend skipping it or putting it in the supplementary.*

AA15: There are several difficulties to conduct in situ cloud measurements. The reason behind our choice to include this discussion as a subsection even though there is no clear dependence between MVD/ED and the position of the probe is to make it clear to the reader that we took into consideration the uncertainties that could be produced from the relative vertical position of the probe with respect to the cloud base altitude. Naturally, some microphysical properties of a cloud are dependent in the air mass origin, while some are determined by the temperature at cloud base and the height of the measurement above cloud base, and some due to the amount of vertical motion and entrainment within the vertical profile. Theoretically, the size parameters ED and MVD are expected to show a dependence on the vertical position of the probe with respect to the cloud base altitude, but they are also dependent on the temperature and initial cloud droplet number concentration at cloud base (Brenguier ,1991). Hence, we wish to show that, while cloud droplet number concentration can clearly be linked to air mass origin, it is more challenging to directly link air mass origin and size parameters (e.g. for comparison with satellite retrievals) without including the cloud microphysical processes happening in the vertical profile. We agree that the clarity of the figure should be improved, and that, in this case, it is not crucial to distinguish between the two instruments. Thus, we created a figure that presents a statistical description of MVD in 5 different altitudes above cloud base. We can see that there was no strong dependency between the vertical position of CAS and FSSP in the cloud and MVD.

[Figure]

Figure 10: Statistical description of hourly averages of median volume diameter (*MVD*) as they were measured by the cloud and aerosol spectrometer (CAS) and the forward-scattering spectrometer probe (FSSP) where PES was within one region >80 % for five different levels of the position of the probes inside the cloud (H) (relative distance of the cloud ground-based spectrometer). Cloud base was measured at the Kenttärova station.

*RC16: l. 428: to be representative or considered as representative*

AA16: The above typo was corrected "considered as representative"

RC17: l. *429: Why are clouds more frequent when air masses originate from Southern and Eastern regions?*

AA17:

It is expected that clouds occurrence depends on the different meteorological conditions that were associated with the different air parcels. The text was modified to answer the reviewer comment.

Line 471 "..regions. This result suggests that clouds occurrence depended on the different meteorological conditions that were associated with the different air parcels. Continental .."

RC18:" *l. 440: What kind of measurements are needed?"*

AA18: We agree with the reviewer that we should elaborate our thoughts. We highlight the need for a bigger amount of cloud measurements that will cover and allow us to investigate a wider temperature range. Particularly, we need to obtain all year around measurements to investigate the temperature dependence. The text was modified.

Instead of "However, more measurements are needed to confirm such temperature dependency of droplet sizes"

To line 494" All year round in situ cloud measurements in the area are of high importance to confirm such temperature dependency of droplet sizes. A larger data set containing a wider temperature range needs to be obtained."

References

Aalto, T., Hatakka, J. and Viisanen, Y. 2003. Influence of air mass source sector on variations in CO2 mixing ratio at a boreal site in northern Finland. Boreal Env. Res. 8, 285–393.

Asmi, E., Kivekäs, N., Kerminen, V.-M., Komppula, M., Hyvärinen, A.-P., Hatakka, J., Viisanen, Y., and Lihavainen, H.: Secondary new particle formation in Northern Finland Pallas site between the years 2000 and 2010, Atmos. Chem. Phys., 11, 12959–12972, https://doi.org/10.5194/acp-11-12959-2011, 2011.

Brenguier, J. L. (1991). Parameterization of the Condensation Process: A Theoretical Approach, Journal of Atmospheric Sciences, 48(2), 264-282.

Eneroth, K., Kjellström, E. and Holmén, K. 2003. A trajectory climatology for Svalbard; investigating how atmospheric flow patterns influence observed tracer concentrations. Phys. Chem. Earth 28, 1191–1203, doi: DOI: 10.1016/j.pce.2003.08.051.

Eneroth, K., Aalto, T., Hatakka, J., Holmen, K., Laurila, T. and Viisanen Y. (2005), Atmospheric transport of carbon dioxide to a baseline monitoring station in northern Finland. Tellus B, 57: 366-374. https://doi.org/10.1111/j.1600-0889.2005.00160.x

Pawlowska, H., W. W. Grabowski, and J. . L. Brenguier: Observations of the width of cloud droplet spectra in stratocumulus. Geophysical Research Letters, 33(19), L19810, doi: 10.1029/2006GL026841, 2006

Pernov, J.B., Beddows, D., Thomas, D.C. *et al.* Increased aerosol concentrations in the High Arctic attributable to changing atmospheric transport patterns. *npj Clim Atmos Sci* **5**, 62 (2022). https://doi.org/10.1038/s41612-022-00286-y

Pisso, I., Sollum, E., Grythe, H., Kristiansen, N. I., Cassiani, M., Eckhardt, S., Arnold, D., Morton, D., Thompson, R. L., Groot Zwaaftink, C. D., Evangeliou, N., Sodemann, H., Haimberger, L., Henne, S., Brunner, D., Burkhart, J. F., Fouilloux, A., Brioude, J., Philipp, A., Seibert, P., and

Stohl, A.: The Lagrangian particle dispersion model FLEXPART version 10.4, Geosci. Model Dev., 12, 4955–4997, https://doi.org/10.5194/gmd-12-4955-2019, 2019.

Seibert, P. and Frank, A.: Source-receptor matrix calculation with a Lagrangian particle dispersion model in backward mode, Atmos. Chem. Phys., 4, 51–63, 2004, SRef-ID: 1680-7324/acp/2004-4-51.

Tunved P., Hansson H.-C., Kerminen V.-M., Ström J., Dal Maso M., Lihavainen H., Viisanen Y., Aalto P.P., Komppula M. & Kulmala M. 2006. High natural aerosol loading over boreal forests. Science 5771, 261–263.2001–2005

---

## Author Comment (AC2)

We sincerely thank reviewer #2 for carefully reading our manuscript, and for their review and constructive comments. We have reviewed the comments and have revised the manuscript accordingly. Our response is given in a point-by-point manner below as reviewer comment (RC) and authors answer (AA).

*In their study Doulgeris et al combine a long-term dataset of in-situ observed cloud microphysical properties at a sub-arctic location with simulations of air mass transport. While the general methodology, uniqueness of the dataset and presentation are reasonable for publication, I do share the concerns of the first referee regarding scientific relevance and quality. In my opinion the manuscript should be reconsidered after major revisions.*

**General comments**

RC1:"*The authors should point out more clearly where their work extends the current level of scientific knowledge. As the authors describe in the literature overview, the Twomey effect is well confirmed, and no significant additions are provided in the manuscript. It should be considered to change the manuscript type and focus to a measurement report instead of a research article.*"

AA1:

The question that is raised is similar as stated by the reviewer 1, accordingly the answers are similar too. We have modified the text to be clearer where this work extends the current level of scientific knowledge. Even though there is an increased demand for long term continuous ground based in-situ cloud measurements, unfortunately there is limited instrumentation available to cover such demand. The atmospheric in-situ measurements community (in our case the European Research Infrastructure for the observation of Aerosol, Clouds and Trace Gases, ACTRIS) has identified cloud droplet probes with surface installation as a potential method for continuous cloud in situ measurements (ACTRIS-PPP Deliverable D5.1: Documentation on technical concepts and requirements for ACTRIS Observational Platforms). However, measurements in conditions like those at our sub-Arctic location are very challenging. To our knowledge, this is the first study that connects extensive in situ cloud measurements to air mass origin. As a result, the dependence of cloud microphysics on the air mass origin in a subarctic mostly pristine region were rarely seen until now using such an in situ long term dataset. We agree with the reviewer that Twomey effect was confirmed however in this work we mainly investigate cloud microphysics and their connection to air mass origin. We point out that there is need of considering not only local meteorological parameters but also the air mass origin in investigations of cloud processes. PaCE measuring period (during autumn) is crucial as it is a unique opportunity to get Arctic pristine air masses (Pernov et al., 2022) and combine them with in situ cloud measurements. Moreover, the procedure of distinguishing cases that correspond to one air mass origin and not to mixed one is complicated and require a huge amount of continuous data. In this work, in situ cloud data with ground-based cloud spectrometers from eight different autumn campaigns were obtained (2004 hours of cloud observations resulted in 706 hours of cloud observations that related to one air mass origin). We proved that cloud microphysical properties and particularly the number concentration of cloud droplets have a strong dependence on the air mass origin. Using those findings, the ACI community can focus on further studies to investigate how aerosol and meteorology of different airmasses along with local meteorological parameters change the cloud microphysics and to what

scale. As a result, we consider this work not just as a measurement report but as a research article that investigating the connection of several microphysical parameters to the cloud origin.

Some of the major changes were applied to abstract, line 22

"…Local). We observed clear differences in the cloud microphysical properties for the air mass source regions. Arctic air masses were characterized by low liquid water content (LWC), low cloud droplet number concentration ($N_c$), and comparatively large median volume and effective droplet diameter. Western region (marine North Atlantic) differed from Arctic by both higher $N_c$ and LWC. Eastern region (continental Eurasia) had only a little higher LWC than Arctic, but substantially higher Nc and smaller droplet diameter. Southern region (continental Europe) had high $N_c$ and LWC, and very similar droplet diameter to the Eastern region. Finally, the relationship between $N_c$ and droplet size (i.e., the Twomey effect) was characterized for the different source regions, indicating that all region clouds were sensitive to increases in $N_c$."

To introduction, line 92 "…at Pallas. To our knowledge, this is the first study that connects extensive in situ cloud measurements to air mass origin. During autumn, clean, natural Arctic background conditions are significantly increasing (Pernov et al., 2022). Subsequently, this allows us to focus in this work on quantifying the impact of air mass origin (e.g., clean arctic vs. long-range transported air from continental Europe) on the microphysical properties of low-level clouds and their patterns based on measurements at the Pallas GAW station. To our knowledge, this is the first study that connects extensive in situ cloud measurements to air mass origin"

To results, line 285 "(a marine environment that the natural Arctic background conditions are significantly increasing (Pernov et al., 2022)"

Line 321 "Averaged temperatures at Sammaltunturi for each air mass were -3.1°C (SD 2.5°C), -2.2 °C (SD 5.9 °C), 1.3 °C (SD 3.9 °C) and -2.8 °C (SD 2.01 °C) for the arctic, eastern, southern and western region respectively. Furthermore, in all regions, there was no clear indication that there was any trend in $N_c$ through different years of PaCEs."

To summary and conclusions line 475, 478 and 488

"This result suggests that clouds occurrence depended on the different meteorological conditions that were associated with the different air parcels"

"The lowest values of cloud droplet concentration were related to clean arctic airmasses. We observed a clear relationship between air mass origin and cloud droplet number concentration. According to theoretical considerations, (Brenguier 1991; Pawlowska et al., 2006) the measurements of cloud droplet number concentration does not depend on the vertical position of the cloud spectrometer within the cloud layer."

"The above differences that were observed in cloud microphysical properties when the air masses were related to different regions show the need to investigate how the aerosol loading and meteorology of different airmasses along with local meteorological parameters change the cloud microphysics and in which scale."

RC2:" *It does not become clear how including the cloud base height information in Sec. 3.4 supports the manuscript. A distance of 4km of the ceilometer for cloud-base height retrieval seems quite far away. Also, it does not become clear if and how only stratiform cases are selected. The resulting Fig 10 looks more like a 'point cloud' without the chance to identify any physical relationship.*"

AA2:

The cloud base height information is used to determine whether cloud events are stratiform or not. The altitude difference between the mountain top station and ceilometer station (210 m) ensures that fog cases are not selected (the liquid cloud base should be above the ground at the ceilometer location) and cloud events must be present at both locations at the same time for a minimum duration of 30 minutes and not varying much in height with time to ensure that the cloud field is stratiform rather than cumulus. The clarity of figure 10 was improved. In the new version, we present a statistical description of MVD in 5 different altitude levels of the cloud. We can see that there was no strong dependency between the vertical position of CAS and FSSP in the cloud and the MVD.

[Figure]

Manuscript Figure 10: Statistical description of hourly averages of median volume diameter (*MVD*) as they were measured by the cloud and aerosol spectrometer (CAS) and the forward-scattering spectrometer probe

(FSSP) where PES was within one region >80 % for five different levels of the position of the probes inside the cloud (H) (relative distance of the cloud ground-based spectrometer). Cloud base was measured at the Kenttärova station.

RC3: *"The airmass source analysis raises some questions as well. The regions seem rather inconsistent. E.g., why is the Kola peninsula 'Eastern' and not 'Arctic' or why is Scotland an Ireland 'Western' while England is 'Southern'. The simulation duration of 4 days is quite short. Was there any sensitivity analysis performed with 7- or 10-day simulations? How were contributions from outside the area of Fig. 3 treated?"*

AA3:

We see this work as a continuation of previous studies that were conducted at Sammaltunturi (e.g., Aalto et.al., 2003, Asmi et al., 2011).  For this reason, we decided to adopt the same source areas, although the borders between the different regions are drawn on a rather coarse scale. Detailed borders of the source regions are given in Table 1 below and included in supplementary information of the revised manuscript; these criteria were used outside the area of Fig. 3. Initially, the regions were classified using trajectories cluster analysis, following the method as Eneroth et al. (2003) proposed. The predefined regions were used for different studies and scopes as atmospheric transport of carbon dioxide (Aalto et.al., 2003, Eneroth et al., 2005), aerosol studies (Tunved et al., 2006; Asmi et al., 2011).

Including Kola peninsula in the Eastern rather than Arctic region ensures that the substantial anthropogenic emissions sources there (e.g., Kyrö et al.,2014) do not mask the remote Arctic air characteristics. As to the Western sector, Ireland, and Scotland (as well as Iceland and Greenland) could be excluded if the areas were redefined. However, the analysis in this manuscript indicates that small contributions (up to 20% of PES, Fig. 6 in manuscript) from other source areas is not critical for the results interpretation. Therefore, we expect that small changes to the borders of the regions would have only a very minor effect on the presented results.

In this analysis, we consider that the transport during the previous 96h is sufficient to classify the air masses into the relatively broad categories. We consider that the requirement of >80% PES within one region during the 96h is a strict criterion. Also, four days period is quite commonly used duration in air mass history analysis for ground-based in-situ measurements (e.g., Asmi et al., 2011; Makonnen et al, 2012, Riuttanen et al., 2013), as aerosols are relatively short-lived in boundary layer. In some cases, such as within the arctic during the polar night or if e.g., long-range transported forest fire smoke is present, longer simulations would be beneficial. However, this is not the case at Pallas during the measurements used here. Therefore, we have not carried out sensitivity analysis with 7- or 10-day simulations but expect that a large portion of the longer air mass history would lie outside of the area in Fig. 3 (see also Fig. 4 example case).

Table 1. Latitude and longitude ranges for each sector.

| Sectors | Latitude (x) | Longitude (y) |
|---|---|---|
| Arctic, marine, area I | $x \geq 70^\circ$ N | |

| | | |
|---|---|---|
| Eastern, continental, area II | x < 70º N | y > 30º E |
| Southern, continental, area III | x < 65º N | 10 < y < 30º E |
| | x < 63º N | 5 < y < 10º E |
| | x < 55º N | 5º W ≤ y < 5º E |
| Western, marine, area IV | 65 ≤ x ≤ 70º N | 10 < y < 15º E |
| | 63 ≤ x ≤ 70º N | 5 < y < 10º E |
| | 55 ≤ x ≤ 70º N | 5º W ≤ y ≤ 5º E |
| | x ≤ 70º N | y ≤ 5º W |
| Local, continental, area V | 65 < x <70º N | 15 < y < 30º E |

Line 224 "…Fig.3. The division was based on previous studies that were conducted at Sammaltunturi (e.g., Aalto et.al., 2003; Eneroth et al., 2005; Tunved et al.,2005, Asmi et al.,2011). Initially, the regions were classified using trajectories cluster analysis, following the method that Eneroth et al. (2003) proposed. The choice of sectors represents roughly the characteristics of the region. The Arctic… "

**Specific comments**

*RC4:" L31: The statement on larger droplets in warm clouds in the current form is not supported by the presented data. Fig 8 b, d shows a decrease of particle size for the 'Arctic' subsample in the FSSP data."*

AA4: In majority of the cases during PaCEs, cloud droplets appeared to be more prone to grow at temperatures larger than –2 °C, however it is true that in Fig 8 b, d there is a decrease of particle size for the 'Arctic' subsample in the FSSP data. This is due to the different amount of data in each temperature bin. For the Arctic region, the observation hours in the last bin were smaller in comparison with the other temperature bins. An explanation was added in the revised manuscript. Also, the number of samples per bin was added in the Supplementary Materials (SM) of the manuscript.

Line 384 "..spectrum. The decrease of particle size for the 'Arctic' subsample in the FSSP data above 0 °C was due to the relatively low amount of observation in this temperature range (2 hours of observation). The observation hours related to each temperature bin for each PaCE are presented in Table 3 of the SM."

Table S3. Observation hours related to temperature bin and each region for all PaCEs.

| Temperature bin ($^0$C) | Arctic(h) | Eastern(h) | Southern(h) | Western(h) |
|---|---|---|---|---|
| (-10,-6) | 32 | 99 | 0 | 0 |
| (-6,-2) | 39 | 85 | 52 | 45 |
| (-2,2) | 45 | 39 | 49 | 59 |
| (2,6) | 2 | 52 | 51 | 14 |
| TOTAL | 118 | 275 | 152 | 118 |

RC5: "*L44: The issue of varying meteorological conditions is raised, but throughout the manuscript is does not become clear how different temperature and humidity within an airmass origin category are treated (or if they are uniform enough to be disregarded*)."

AA5: Average temperatures per region as measured in situ are provided below and were added in the manuscript (line 318). In the presented dataset, temperature range per region is not wide enough to notice crucial changes in the microphysical properties of the cloud and air mass origin seem to be the most crucial parameter. $N_c$ was not strongly dependent on temperature in this dataset. MVD and ED had minor changes (less than 1um) in several cases, e.g., eastern region from -8 to 0 $^\circ$C. The dependence of ED and MD on temperature was discussed in section 3.3. Relative humidity values measured at the station during cloud event were always approximately 100 %. Meteorological parameters at different altitudes at the Sammaltunturi station were also analyzed using ERA 5 re analysis and the results are provided to SM. Based on ERA5 profiles T and Q variability is smallest in Arctic air mass. The Eastern and Western air mass medians are close to the Arctic air mass, but the Southern sector has clearly higher T and Q.

[Figure]

Figure S1. ERA5 temperature(T), specific humidity(Q) and wind speed (WS)profiles for the cases, when at least 80% of PES was within a source region. Line is the median and error bars indicate upper and lower quartiles. I, II, III and IV corresponds to the arctic, eastern, southern, and western region respectively. Station pressure is ~970 hPa.

Line 321 "Averaged temperatures at Sammaltunturi for each air mass were -3.1°C (SD 2.5°C), -2.2 °C (SD 5.9 °C), 1.3 °C (SD 3.9 °C) and -2.8 °C (SD 2.01 °C) for the arctic, eastern, southern and western region respectively."

*RC6: "L49-51: Consider rephrasing this sentence, it is hard to grasp what is reason of the limited knowledge and what is the consequence."*

AA6: Text was modified as the reviewer suggested.

Line 52 "It is important to understand how different air masses can influence the aerosols and the cloud microphysics when the cloud dynamics and the interaction between aerosols and clouds are examined (e.g., Painemal et al., 2014; Orbe et al., 2015a; Fuchs et al., 2017; Cho et al., 2021)."

*RC7: " L79-83: The sudden appearance of ice particle sizes confuses the reader. As the manuscript focuses only on cloud droplets, consider removing it"*

AA7: The introduction is focusing on clouds and air mass origin. This work is one the few studies that investigate long term cloud properties so we would like to keep this reference in our introduction. However, as the reviewer suggested we will modify the text to avoid any readers' confusion.

*RC8: "Fig 1: The information content of this map has to be increased. Include the elevation, ideally as shading or contour line, as you later argue based on the orography. A legend and lat-lon grid are lacking. Does the darker green color indicate forest? The labels are too small."*

AA8: The above suggestion was accepted; A new map was created.

*RC9: "L147 and 180: Please provide a histogram of wind direction/wind strength. Are certain airmass origin categories subsampled due to filtering periods when FSSP and CAPS did not look into the same direction."*

AA9: There were not any data subsampled in the period when FSSP and CAS did not look into the same direction. As we highlight in line 181 "We only used measurements when the cloud spectrometers were facing the wind direction".. This is the main reason that throughout the manuscript we present the two instruments separately. The obtained data set from the two instrument setups is different due to their different operational times (see table 2). The histogram of the wind direction for the CAS probe will not provide any further information as we used data when the instrument was looking to the wind direction (225 +- 25). The histogram of the wind direction for the FSSP data set is provided below. Wind speed was in all cases lower than the probe air speed of both setups (for the FSSP 6.7 m/s (SD 2.4 m/s) and for the CAS 7.1 m/s (SD 2.3 m/s)). A detailed description of all PaCEs, both ground setups, installation, limitations and the methodology that was used is documented in previous studies (Doulgeris et al., 2020, Doulgeris et al.,2022).

[Figure]

Figure 1. Wind direction histogram for the FSSP data set that was used in this work.

RC10: *" L152-157: At which of the sites were the meteorological observations conducted? Sammaltunturi? Given the amount of detail on hardware in this paragraph, it would be nice to have that the reader would not be forced to consult Doulgeris 2020/2022 for this piece of information."*

AA10: Meteorological observations were conducted at the Sammaltunturi site. Text was modified as the reviewer suggested.

Line 156 "… that was deployed at the Sammaltunturi site. All.."

RC11: "*Fig 6 b: Please distinguish the years. Make clearer what data is from CAS and what from FSSP*"

The clarity of figure 6b was improved, years were added. Thus, fig. 5b was replaced by fig.7 of the revised manuscript.

[Figure]

Manuscript figure 7: Cloud droplet number concentration ($N_c$) for each region and single PaCE campaign as they were measured by the cloud and aerosol spectrometer (CAS) and the forward-scattering spectrometer probe (FSSP) where the PES fraction was within one region >80 % (full symbols) and the PES fraction was within one region from 70 to 80 % (open symbols). Error bars indicate the corresponding standard deviation.

*RC12: "L304: Please provide the duration of the >80% periods also as fraction of the total in-cloud duration"'*

AA12: In total 2004 hours of cloud observations resulted in 706 hours of cloud observations that related to non-mixed air mass origins. The following information was added

Line 322: " ..probe (from total 2004 hours of in situ cloud data 706 hours belongs to non-mixed air mas origin.)"

RC13: *"Fig 7: Without indicating the variability, the yearly averages are of limited use for the reader. How may hours of observations are available for each year and each cluster? Also, given the typical duration of the campaigns, shown is an autumn average."*

AA13: Variability information were added in the Supplementary Materials (SM) of the revised manuscript, for each year and region. Figure 7 was updated and simplified. We also agree that "yearly average" is misleading the reader, and it was changed to PaCEs average.

[Figure]

Figure 8. Cloud droplet size distribution associated with the (a) Arctic, (b) Western, (c) Southern and (d) Eastern region as they were measured by the cloud and aerosol spectrometer (CAS) and the forward-scattering spectrometer probe (FSSP) during all PaCEs.

Table S2. In total 706 observation hours of non-mixed airmasses related to each region for each PaCE.

|  | Arctic(h) | Eastern(h) | Southern(h) | Western(h) |
|---|---|---|---|---|
| 2005 | 0 | 0 | 11 | 0 |
| 2009 | 0 | 0 | 29 | 9 |
| 2012 | 30 | 53 | 0 | 10 |
| 2013 | 22 | 54 | 16 | 25 |
| 2015 | 8 | 138 | 58 | 46 |
| 2017 | 18 | 30 | 0 | 28 |
| 2019 | 40 | 0 | 38 | 0 |
| TOTAL | 118 | 275 | 152 | 118 |

*RC14: "L350-352: The statement on shorter lifetime of warm Arctic clouds not well supported by the data presented. Please either extend the argumentation or omit that senstence."*

AA14: Sentence was omitted as the reviewer suggested.

*RC15: "Fig 8: Please include the no of samples per bin, instead the vague statement in L354."*

AA15: The number of samples per bin was added in the SM of revised manuscript.

Table S3. Observation hours related to temperature bin and each region for all PaCEs.

| Temperature bin ($^0$C) | Arctic(h) | Eastern(h) | Southern(h) | Western(h) |
|---|---|---|---|---|
| (-10,-6) | 32 | 99 | 0 | 0 |
| (-6,-2) | 39 | 85 | 52 | 45 |
| (-2,2) | 45 | 39 | 49 | 59 |
| (2,6) | 2 | 52 | 51 | 14 |
| TOTAL | 118 | 275 | 152 | 118 |

RC16: *"L367: How did you account for different temperatures in different air masses for this conclusion?"*

AA16: This finding is indeed attributed to the typical temperatures at cloud base for each air mass origin, as the temperature at cloud base determines the gradient of LWC with height (Brenguier 1991). Temperatures at cloud base were not directly obtained but the temperature at the Sammaltunturi measurement station provides a good estimation. Air masses with colder temperatures showed lower LWC values, and the air mass with the broadest temperature range also showed a wider range of LWC values.

A sentence has been added earlier in the manuscript at Line 318: "Averaged temperatures at Sammaltunturi for each air mass were -3.1°C (SD 2.5°C), -2.2 °C (SD 5.9 °C), 1.3 °C (SD 3.9 °C) and -2.8 °C (SD 2.01 °C) for the arctic, eastern southern and western region respectively."

RC17*: Fig 9 a: similarly to Fig 6b, please indicate the single years and make CAPS and FSSP more distinguishable*

AA17: In the revised manuscript after updating Fig. 7 (or previously 6b) we show that there is not a clear yearly dependence in $N_c$. Thus, we proved that that there were no obvious changes through years or possible malfunction of the instruments that were used and could produce biased results. So, we decided that the best option would be to simplify Figure 9a (figure 10a in latest version) and include only the statistical description.

[Figure]

Figure 9: Statistical description of liquid water content (LWC) for each region as it was measured by the cloud and aerosol spectrometer (CAS) and the forward-scattering spectrometer probe (FSSP) where PES was within one region >80 %.

RC18: "*L431-434: The conclusions on number concentration and diameter with respect to the height are not backed by the analysis presented in Sec 3.4 at all. Either remove this aspect or expand on the reasoning, including descriptive figures.*"

AA18: We have rewritten this statement. What we wanted to highlight in the conclusion is that, according to theory, the measurement of cloud droplet number concentration should not depend on the height into cloud at which the measurement is made (Pawlowska et al., 2006), therefore relationships between air mass origin and cloud droplet number concentration should be possible to observe. Such relationships are not expected between air mass origin and droplet size (Brenguier 1991; Boers et al., 2000) as these require additional considerations (temperature at cloud base, cloud droplet number concentration, rate of increase of LWC with height, height of measurement above cloud base).

Replace: "Number concentration was expected to be a robust signal with no dependency from the vertical position of cloud spectrometer in the cloud. On the other hand, both effective radius and median volume diameter has some extra uncertainties depending on the altitude with respect to cloud base."

with line 482 "We observed a clear relationship between air mass origin and cloud droplet number concentration. According to theoretical considerations (Brenguier 1991; Pawlowska et al., 2006), the measurements of cloud droplet number concentration does not depend on the vertical position of the cloud spectrometer within the cloud layer."

References

Aalto, T., Hatakka, J. and Viisanen, Y. 2003. Influence of air mass source sector on variations in $CO_2$ mixing ratio at a boreal site in northern Finland. Boreal Env. Res. 8, 285–393.

Asmi, E., Kivekäs, N., Kerminen, V.-M., Komppula, M., Hyvärinen, A.-P., Hatakka, J., Viisanen, Y., and Lihavainen, H.: Secondary new particle formation in Northern Finland Pallas site between the years 2000 and 2010, Atmos. Chem. Phys., 11, 12959–12972, https://doi.org/10.5194/acp-11-12959-2011, 2011.

Brenguier, J. L. (1991). Parameterization of the Condensation Process: A Theoretical Approach, Journal of Atmospheric Sciences, 48(2), 264-282.

Eneroth, K., Kjellström, E. and Holmén, K. 2003. A trajectory climatology for Svalbard; investigating how atmospheric flow patterns influence observed tracer concentrations. Phys. Chem. Earth 28, 1191–1203, doi: DOI: 10.1016/j.pce.2003.08.051.

Eneroth, K., Aalto, T., Hatakka, J., Holmen, K., Laurila, T. and Viisanen Y. (2005), Atmospheric transport of carbon dioxide to a baseline monitoring station in northern Finland. Tellus B, 57: 366-374. https://doi.org/10.1111/j.1600-0889.2005.00160.x

Kyrö, E.-M., Väänänen, R., Kerminen, V.-M., Virkkula, A., Petäjä, T., Asmi, A., Dal Maso, M., Nieminen, T., Juhola, S., Shcherbinin, A., Riipinen, I., Lehtipalo, K., Keronen, P., Aalto, P. P., Hari, P., and Kulmala, M.: Trends in new particle formation in eastern Lapland, Finland: effect of decreasing sulfur emissions from Kola Peninsula, Atmos. Chem. Phys., 14, 4383–4396, https://doi.org/10.5194/acp-14-4383-2014, 2014Aalto, T., Hatakka, J. and Viisanen, Y. 2003. Influence of air mass source sector on variations in $CO_2$ mixing ratio at a boreal site in northern Finland. Boreal Env. Res. 8, 285–393.

Makkonen, U., Virkkula, A., Mäntykenttä, J., Hakola, H., Keronen, P., Vakkari, V., and Aalto, P. P.: Semi-continuous gas and inorganic aerosol measurements at a Finnish urban site: comparisons with filters, nitrogen in aerosol and gas phases, and aerosol acidity, Atmos. Chem. Phys., 12, 5617–5631, https://doi.org/10.5194/acp-12-5617-2012, 2012.

Pawlowska, H., W. W. Grabowski, and J. . L. Brenguier: Observations of the width of cloud droplet spectra in stratocumulus. Geophysical Research Letters, 33(19), L19810, doi: 10.1029/2006GL026841, 2006

Pernov, J.B., Beddows, D., Thomas, D.C. *et al.* Increased aerosol concentrations in the High Arctic attributable to changing atmospheric transport patterns. *npj Clim Atmos Sci* **5**, 62 (2022). https://doi.org/10.1038/s41612-022-00286-y

Riuttanen, L., Hulkkonen, M., Dal Maso, M., Junninen, H., and Kulmala, M.: Trajectory analysis of atmospheric transport of fine particles, $SO_2$, $NO_x$ and $O_3$ to the SMEAR II station in Finland in 1996–2008, Atmos. Chem. Phys., 13, 2153–2164, https://doi.org/10.5194/acp-13-2153-2013, 2013

Tunved P., Hansson H.-C., Kerminen V.-M., Ström J., Dal Maso M., Lihavainen H., Viisanen Y., Aalto P.P., Komppula M. & Kulmala M. 2006. High natural aerosol loading over boreal forests. Science 5771, 261–263.2001–2005